# WDFY2 restrains matrix metalloproteinase secretion and cell invasion by controlling VAMP3-dependent recycling

Marte Sneeggen[1,2], Nina Marie Pedersen[1,2], Coen Campsteijn[3], Ellen Margrethe Haugsten[1,4], Harald Stenmark[1,2] & Kay Oliver Schink [1,2]

Cancer cells secrete matrix metalloproteinases to remodel the extracellular matrix, which enables them to overcome tissue barriers and form metastases. The membrane-bound matrix metalloproteinase MT1-MMP (MMP14) is internalized by endocytosis and recycled in endosomal compartments. It is largely unknown how endosomal sorting and recycling of MT1-MMP are controlled. Here, we show that the endosomal protein WDFY2 controls the recycling of MT1-MMP. WDFY2 localizes to endosomal tubules by binding to membranes enriched in phosphatidylinositol 3-phosphate (PtdIns3P). We identify the v-SNARE VAMP3 as an interaction partner of WDFY2. WDFY2 knockout causes a strong redistribution of VAMP3 into small vesicles near the plasma membrane. This is accompanied by increased, VAMP3-dependent secretion of MT1-MMP, enhanced degradation of extracellular matrix, and increased cell invasion. WDFY2 is frequently lost in metastatic cancers, most predominantly in ovarian and prostate cancer. We propose that WDFY2 acts as a tumor suppressor by serving as a gatekeeper for VAMP3 recycling.

---

[1] Centre for Cancer Cell Reprogramming, Institute of Clinical Medicine, Faculty of Medicine, University of Oslo, Montebello N-0379 Oslo, Norway. [2] Department of Molecular Cell Biology, Institute for Cancer Research, Oslo University Hospital, Montebello N-0379 Oslo, Norway. [3] Department of Molecular Medicine, Institute of Basic Medical Sciences, University of Oslo, 0317 Oslo, Norway. [4] Department of Tumor Biology, Institute for Cancer Research, Oslo University Hospital, Montebello, Oslo N-0379, Norway. Correspondence and requests for materials should be addressed to H.S. (email: h.a.stenmark@medisin.uio.no) or to K.O.S. (email: Kay.Oliver.Schink@rr-research.no)

One of the most life-threatening aspects of cancer is the ability of transformed cells to invade into the extracellular matrix (ECM) and neighboring tissue to form metastases[1,2]. Metastasis is correlated with aggressive tumors and poor prognosis for the patient and therefore one of the leading causes of death by cancer[1]. Matrix metalloproteinases (MMPs) play a critical role in progression of cancer by degrading and remodeling ECM, enabling cells to overcome tissue barriers, travel within the circulatory system before extravasating to produce a secondary tumor[3,4]. Membrane-bound MMPs such as MT1-MMP (also called MMP14) are internalized by clathrin-dependent and caveolar endocytosis[5]. After internalization, MT1-MMP is sorted in endosomal compartments, and a fraction is recycled back to the plasma membrane. How MMPs are sorted in endosomes is largely unknown. Sorting of endocytic cargos occurs at specialized tubular domains of early endosomes. It is likely that sorting of MMPs occurs by similar mechanisms, however, the molecular factors that regulate this process are largely unknown.

Here, we have studied the function of WDFY2, a cytosolic protein that has been described to reside on endocytic vesicles close to the plasma membrane[6]. It contains a lipid-binding FYVE domain and seven WD40 repeats which can form a β-propeller. Potentially, β-propellers can act as platforms for protein–protein interactions, but only few interactors of WDFY2 have been identified to date, and its cellular functions remain to be elucidated[7,8].

We show that WDFY2 regulates exocytosis of MT1-MMP by controlling endosomal sorting of the v-SNARE VAMP3. WDFY2 localizes to actin-stabilized endosome tubules positive for the small GTPase RAB4 and shows a preference for highly curved membranes enriched for the lipid phosphatidylinositol 3-phosphate (PtdIns3P). Here, it interacts with VAMP3, which directs secretion of endosome-derived cargos, including MT1-MMP. We show that loss of WDFY2 leads to enhanced secretion of MT1-MMP and allows cells to actively invade into ECM.

## Results

**WDFY2 localizes to early endosomes.** After internalization and uncoating, endocytic vesicles gain the early endocytic marker APPL1. APPL1 vesicles can mature into WDFY2-positive vesicles. These vesicles can then further mature into early endosomes containing the canonical marker EEA1[9]. To date, the function of WDFY2 in the endocytic pathway is poorly characterized. To define the localization of WDFY2 in the endocytic pathway, we transiently transfected hTERT-RPE1 cells with green flourescent protein (GFP)-WDFY2 and performed structured illumination microscopy (SIM) together with APPL1 and EEA1 visualized with antibodies (Fig. 1a). Whereas APPL1- positive vesicles localize close to the plasma membrane, we observed that WDFY2 localized to a pool of vesicles that was further from the plasma membrane and negative for APPL1. WDFY2 showed labeling of two vesicle pools, a small pool negative for both APPL1 and EEA1, and one major pool, which is positive for EEA1. Notably, on endosomes positive for EEA1, WDFY2 did not completely colocalize with EEA1, but rather localized to distinct, EEA1-negative subdomains (Fig. 1a).

Earlier reports have proposed that WDFY2 does not localize to endosomes positive for the early endosomal marker RAB5, and therefore it has been suggested that WDFY2 marks a different set of endosomes which is distinct from those enriched in EEA1[6,9]. We therefore asked if WDFY2 localized together with any of the well-characterized RAB GTPases in the early endocytic pathway. SIM was performed using a stable cell line expressing GFP-WDFY2, which was transiently transfected with mCherry-RAB4,

-RAB5, and -RAB11. Endogenous RAB7 was visualized by antibody staining (Fig. 1b). This imaging showed that WDFY2, in contrast to previous reports, localized to endosomes positive for RAB5, but resided on distinct endosomal subdomains with only partial overlap with RAB5 (Fig. 1b). We also observed that WDFY2 localized to endosomal regions positive for the fast-recycling marker RAB4. In contrast, we observed less overlap with the late-endosomal marker RAB7, and we could not observe colocalization with RAB11, a marker for the slow recycling pathway (Fig. 1b).

Manders' colocalization analysis on confocal images confirmed colocalization of WDFY2 with RAB4 and RAB5, whereas there was only limited colocalization with RAB7 and RAB11 (Fig. 1c).

To exclude the possibility that WDFY2 mislocalizes due to overexpression, we generated an endogenously N-terminal localization and purification (NLAP)-tagged WDFY2 allele using CRISPR/Cas9 and AAV-based donor delivery in hTERT-RPE1 cells (Supplementary Fig. 1a, c). Resulting cell lines were stained with antibodies against GFP and EEA1 and analyzed by confocal microscopy (Fig. 1d). We found that also WDFY2 expressed at endogenous levels, similar to stable WDFY2 cell lines, localized as subdomains on EEA1-positive endosomes. Taken together, our data show that WDFY2 labels subdomains on early endosomes. The colocalization with RAB4 on these structures indicates that WDFY2 localizes to membrane structures that regulate recycling.

**WDFY2 is enriched on actin-stabilized endosomal tubules.** To analyze the dynamic localization of WDFY2, we performed live-cell microscopy using the stable cell line expressing GFP-WDFY2 and the endogenously-tagged NLAP-WDFY2 cell lines to trace the localization of WDFY2 in the endocytic pathway. Time-lapse movies showed GFP-WDFY2 localizing to endosomes, with a prominent localization to tubulating regions. These tubules emerged from WDFY2-rich domains on the endosomal membrane and showed strong accumulation of WDFY2 (Fig. 2a, Supplementary Movie 1). To test if also endogenous WDFY2 localizes to tubular structures or if the observed tubulation is caused by overexpression of WDFY2, we used cells expressing endogenously-tagged WDFY2. These were transfected with mCherry-tagged EEA1 and analyzed by fluorescence microscopy (Fig. 2d). We observed that endogenously-tagged WDFY2 selectively localized to endosomal tubules originating from EEA1-positive endosomes (Fig. 2d, Supplementary Movie 2). In comparison to stable cell lines expressing GFP-WDFY2, the tubular localization was more pronounced and only limited localization to the limiting membrane of the endosome could be observed. In order to gain super-resolved images of the tubular structures, we performed stochastic optical reconstruction microscopy (STORM) on fixed cells stably expressing GFP-WDFY2. This imaging showed an accumulation of WDFY2 at tubules emerging from the otherwise round endosome, as well as accumulations of WDFY2 at the base of these tubules (Fig. 2b). In contrast, the remaining limiting membrane of the endosomes showed only weak WDFY2 staining. In addition, we performed DNA points accumulation for imaging in nanoscale topography (DNA-PAINT) microscopy of GFP-WDFY2 together with HRS, a protein involved in degradative protein sorting on early endosomes[10,11]. WDFY2 labels both the limiting membrane and endosomal tubules, whereas HRS localized only to microdomains of the vesicles and could not be detected on tubules positive for WDFY2 (Fig. 2c).

Localization of WDFY2 to endosomal tubules has not been reported before and we therefore proceeded to characterize these structures in detail. Endosomes give rise to different classes of tubules which can mediate sorting and transport of cargos.

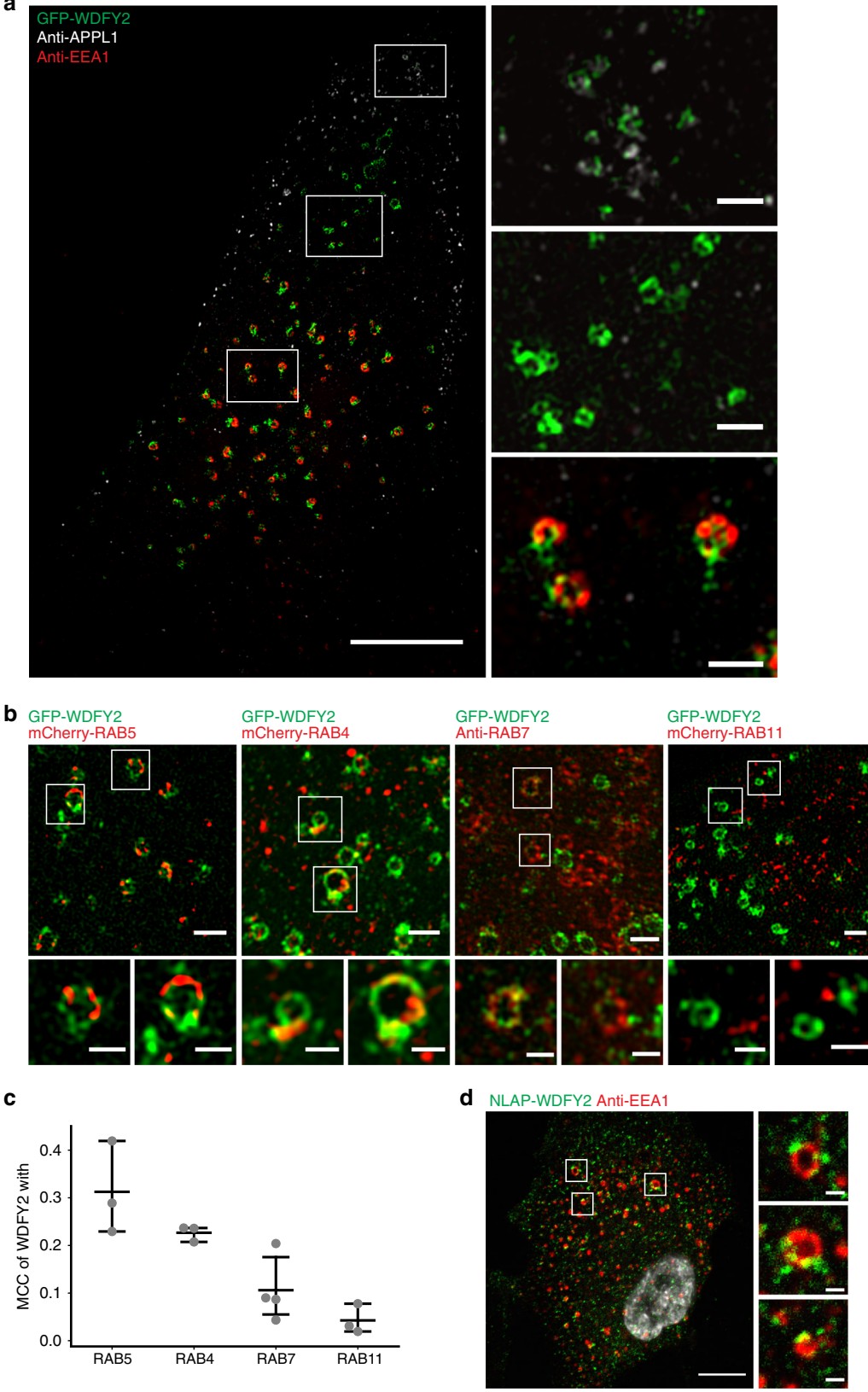

Previous reports showed that actin-dependent, relatively stable tubules are involved in the sorting of slowly diffusing cargos such as β2-adrenergic receptors, whereas bulk recycling of freely diffusing cargo such as transferrin receptor (TFR) happens via short-lived tubules[12]. We therefore investigated to which tubule class WDFY2 localizes. We found that GFP-WDFY2-labeled tubules were relatively long-lived and showed accumulations of the actin organizing proteins CORONIN1B and WASH at their base (Fig. 2e, f, h and Supplementary Movie 3)[13,14]. In addition, the WASH-interacting protein FAM21 and subunits of the

**Fig. 1** WDFY2 localization in the endocytic pathway. **a** SIM image showing representative WDFY2 localization in relation to different markers of the early endocytic pathway. APPL1 (gray) was used as a marker for early vesicles and EEA1 (red) marks early endosome. GFP-WDFY2 (green) localizes to an independent vesicle pool (inset 2) and to subdomains on EEA1-labeled endosomes (inset 3). There is only limited overlap with APPL1 endosomes (inset 1). Scale bar: 10 μm, insets 1 μm. Representative image of 14 cells. **b** SIM images showing the localization of GFP-WDFY2 in relation to mCherry-RAB5, mCherry-RAB4, RAB7 or mCherry-RAB11. Shown are representative images of 12(RAB5), 13(RAB4), 14(RAB7, RAB11) cells. Scale bar: 1 μm, insets: 0.5 μm. **c** Colocalization using Manders' colocalization coefficient (MCC) analysis of hTERT-RPE1 cells stably expressing GFP-WDFY2 shows overlap between WDFY2 and RAB4 and RAB5, but only moderate overlap with RAB7 and RAB11. $n = 3$ experiments (Rab4, Rab5, and Rab11) and $n = 4$ (Rab7) with 70 cells per condition. Shown are individual experiments and the mean ± 95% CI. **d** Confocal image showing localization of endogenously-tagged WDFY2 (green) and EEA1 (red). Scale bar: 10 μm, insets: 1 μm. Representative image from 14 cells. Source data are provided as a Source Data file

retromer cargo recognition complex—VPS26 and VPS35 (Fig. 2g, h and Supplementary Fig. 2a, b)[14,15]—localized to the base of GFP-WDFY2-labeled tubules.

In order to assess the role of actin in the formation and stabilization of WDFY2-labeled tubules, we treated cells expressing GFP-WDFY2 with the actin depolymerizing drug LatrunculinB and the ARP2/3 inhibitor CK666. Depolymerization of actin with Latrunculin caused rapid changes in the morphology of GFP-WDFY2-labeled tubules. We observed that, after addition of LatrunculinB, endosomal tubules started to hyper-tubulate (mean length: 1.1 μm (LatrunculinB) vs. 0.74 μm (control)) and pinched off as large fragments (Supplementary Fig. 2c, e and Supplementary Movie 4). Endosomes in cells treated with the ARP2/3 inhibitor CK666 also showed elongated tubules (mean length: 1.33 μm (CK666) vs. 0.85 μm (Control)) (Supplementary Fig. 2d, e) after addition of the inhibitor. This suggests that WDFY2 localizes to actin-stabilized tubules which are involved in specific sorting of cargo. Together with the colocalization with RAB4, this suggests that WDFY2 could be involved in controlling recycling of endocytosed cargos.

**The PtdIns3P-binding FYVE domain mediates WDFY2 localization.** We then investigated what drives localization of WDFY2 to endosomal subdomains. WDFY2 contains a FYVE domain and a predicted β-propeller. Whereas most FYVE domains bind PtdIns3P, also β-propellers have been described to be potential phosphoinositide (PI) binders[6]. We therefore tested if full-length WDFY2 was able to bind PtdIns3P and other PIs. By purifying full-length WDFY2 and performing a protein–lipid overlay assay, we found that WDFY2 selectively binds to PtdIns3P (Fig. 3a). To test if this specificity is governed by the FYVE domain, we generated a stable cell line expressing GFP-WDFY2 with a point mutation in the conserved binding site for PtdIns3P (R315A), which abolishes PtdIns3P binding without distorting the overall FYVE structure[16,17]. This mutation completely disrupted the localization of WDFY2 to endosomes and led to a cytosolic localization (Fig. 3b). This indicates that binding of the FYVE domain to PtdIns3P is critical for the endosomal localization of WDFY2.

To further support the role of PtdIns3P for WDFY2 recruitment, we used inhibitors against PI 3-kinases, the enzymes that produce 3-PIs. Addition of either the general PI 3-kinase inhibitor Wortmannin or SAR405, a highly specific inhibitor of the PtdIns3P-producing PI 3-kinase VPS34, led to rapid loss of WDFY2 from endosomes (Fig. 3c, Supplementary Fig. 3)[18,19]. Thus, we conclude that WDFY2 is recruited to endosomes by FYVE-dependent PtdIns3P binding.

Next, we tested if the tubular localization of WDFY2 is mediated by the PtdIns3P-binding FYVE domain or other features of WDFY2. To this end, we generated a tandem PtdIns3P-binding probe based on the FYVE domain of WDFY2 (2xFYVE) and compared it to the widely used HRS-derived 2xFYVE probe[20]. Surprisingly, whereas HRS 2xFYVE localized only weakly to tubular structures, the WDFY2-derived 2xFYVE probe showed a strong preference for tubular structures (Fig. 3d).

We then asked why these two PtdIns3P-binding domains showed this divergent subcellular localization. An attractive hypothesis is that the WDFY2 FYVE domain shows curvature-dependent binding to PtdIns3P-containing membranes and thereby preferentially localizes to highly curved tubular structures. To test this hypothesis, we performed liposome flotation assays using PtdIns3P-containing liposomes extruded to different diameters[21]. These assays showed that the WDFY2-2xFYVE domain preferentially binds to liposomes with high curvature, whereas the HRS-2xFYVE domain does not show any clear preference (Fig. 3e, f). Taken together, our data suggest that localization of WDFY2 to small vesicles and endosomal tubules is controlled by its FYVE domain. Moreover, they highlight that different FYVE domains, despite binding to the same PI, can show preferences for distinct lipid subpopulations, based on the physical properties of the PI-containing membrane.

**WDFY2 interacts with the v-SNARE VAMP3.** Little is known about the function of WDFY2 in the endocytic pathway, and few WDFY2 interaction partners are described so far[7,8]. To further elucidate the function of WDFY2, we set out to identify putative interaction partners. GFP-Trap immuno-precipitates from cell lines stably expressing GFP-WDFY2 or GFP as a control were analyzed by quantitative mass spectrometry (Supplementary Data 1).

Using this method, we identified the v-SNARE protein VAMP3 as a potential interaction partner, which showed 52-fold enrichment in the mass spectrometry analysis (Fig. 4a). VAMP3 has been shown to be present on recycling endosomes and involved in Transferrin recycling[22]. It can also bind to the plasma membrane through the t-SNAREs SYNTAXIN1, SYNTAXIN4, SNAP23, and SNAP25[23]. VAMP3 is usually segregated into tubular membranes where it facilitates fusion with the endocytic compartment and the Golgi apparatus[22]. To verify this interaction, we performed GFP affinity purifications followed by western blotting. This experiment confirmed the interaction of WDFY2 and VAMP3 (Fig. 4b).

Next, we asked whether WDFY2 and VAMP3 colocalize on the same vesicles and if VAMP3 resides on WDFY2-positive tubules. To this end, we performed SIM imaging of cells transiently transfected with GFP-VAMP3 and mCherry-WDFY2 (Fig. 4c). This revealed that WDFY2 and VAMP3 were localized to the same endosomes. In addition, VAMP3 could also be found on WDFY2 negative vesicles (Fig. 3c). Live-cell imaging showed that WDFY2 and VAMP3 colocalized at endosomal tubules and VAMP3-labeled vesicles arising from these tubules (Supplementary Fig. 4).

In order to follow VAMP3 and WDFY2 dynamics in more detail, we used VAMP3 tagged with the photoactivatable fluorescent reporter PA-mCherry. Selective photoactivation of individual endosomes showed that VAMP3 preferentially localized to WDFY2-positive tubules (Fig. 4d, Supplementary

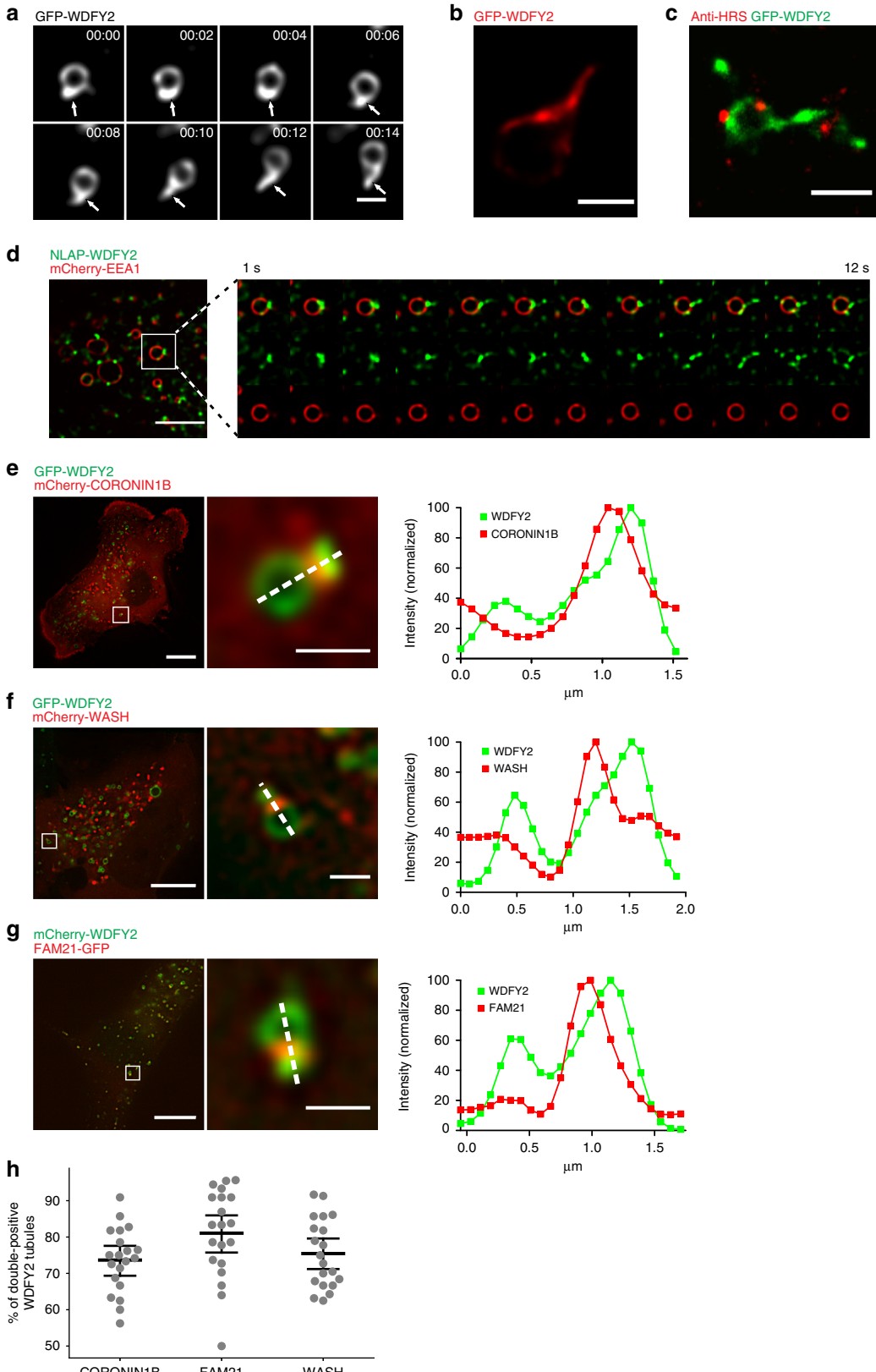

Movie 5). From these tubules, vesicles positive for VAMP3 and WDFY2 were pinched off.

**VAMP3 accumulates at the leading edge of WDFY2 knockout cells**. In order to analyze the biological function of WDFY2, we

generated a hTERT-RPE1 WDFY2 knockout cell line using CRISPR/Cas9 (Supplementary Fig. 5a, b)[24]. We used these cells to address whether VAMP3 localization is affected by the absence of WDFY2.

In hTERT-RPE1 cells, VAMP3 preferentially localized to vesicles which were dispersed throughout the cell and showed

**Fig. 2** WDFY2 localizes to endosomal tubules. **a** Sequential images showing GFP-WDFY2 localization to newly formed tubular structures and GFP-WDFY2 accumulation on the base of the tubules. Shown are frames from a timelapse sequence with images acquired every 2 s. Representative image from 200 observed endosomes. Scale bar: 0.5 μm. **b** dSTORM image showing GFP-WDFY2 localization to tubules. Representative image of five cells. Scale bar: 0.5 μm. **c** DNA-PAINT image showing GFP-WDFY2 and HRS localization to endosomes. Representative image of three cells. Scale bar: 0.5 μm. **d** Sequential images of endogenously NLAP-tagged WDFY2 showing localization to endosomes positive for mCherry-EEA1 and endosomal tubules. Shown are images from a time-lapse movie with images acquired every second. Representative image from 31 movies. Scale bar: 5 μm. **e** Localization of GFP-WDFY2 and mCherry-CORONIN1B on a tubulating endosome. CORONIN1B localizes to the base of WDFY2-positive tubules. The graph shows the normalized fluorescence intensity of GFP-WDFY2 and mCherry-CORONIN1B along the indicated line. Representative image from 50 observed endosomes. Scale bar 10 μm/1 μm (inset). **f** Localization of GFP-WDFY2 and mCherry-WASH on a tubulating endosome. WASH localizes to the base of WDFY2-positive tubules. The graph shows the normalized fluorescence intensity of GFP-WDFY2 and mCherry-WASH along the indicated line. Representative data from 50 observed endosomes. Scale bar 10 μm/1 μm (inset). **g** Localization of mCherry-WDFY2 and GFP-FAM21 on a tubulating endosome. FAM21 localizes to the base of WDFY2-positive tubules. The graph shows the normalized fluorescence intensity of mCherry-WDFY2 and GFP-FAM21 along the indicated line. Representative data from 50 observed endosomes. Scale bar 10 μm/1 μm (inset). **h** Quantification of WDFY2 tubules containing CORONIN1B, WASH, or FAM21. Results are shown as percentage of WDFY2 tubules positive for CORONIN1B, WASH or FAM21 per cell (20 cells per condition, CORONIN1B = 332 tubules, WASH = 338 tubules, FAM21 = 309 tubules). Plotted are the percentage of double-positive tubules per cell and the mean ± 95% CI. Source data are provided as a Source Data file

an accumulation at vesicles in the area surrounding the Golgi apparatus (Fig. 5a, Supplementary Fig. 5c). To study the effect of WDFY2 on the distribution of VAMP3, hTERT-RPE1 wild-type (WT) cells, and hTERT-RPE1 WDFY2 knockout (WDFY2 (−/−)) cells were transfected with GFP-VAMP3. We observed that in knockout cells, GFP-VAMP3 vesicles clustered just beneath the leading edge of migrating cells, which was labeled with mCherry-CORTACTIN (Fig. 5a). In WT cells, this localization was not as prominent. VAMP3 is evenly distributed in WT cells, with small peaks close to the nucleus and the leading edge (Fig. 5b, Supplementary Fig. 9a). In comparison, WDFY2 (−/−) cells showed a dramatic redistribution of VAMP3, with low concentrations of VAMP3 localized close to the nucleus and a strong accumulation of VAMP3 close to the leading edge of the cell.

**WDFY2 controls VAMP3-dependent secretion.** We next asked if this accumulation of vesicles at the leading edge could result in higher rates of VAMP3-driven endocytic recycling to the plasma membrane. To this end, we utilized a pHluorin-based exocytosis assay[25]. We transfected WT and WDFY2(−/−) cells with VAMP3-pHluorin. In cells expressing this construct, the pH-sensitive pHluorin tag faces the lumen of endocytic and secretory vesicles and is exposed at the cell surface as the vesicle fuses with the plasma membrane. In the acidic environments of endosomes, the pHluorin fluorophore is quenched. Once the recycling vesicle has fused with the plasma membrane, the fluorophore is unquenched. Cells were imaged every second for 2 min using total internal reflection fluorescence imaging (TIRF) and bright flashes, indicating fusion events within the plasma membrane, were counted manually using Fiji[26]. We observed that the recycling rate was significantly elevated in the WDFY2(−/−) cells compared to the WT cells, indicating that loss of WDFY2 does not only lead to redistribution of VAMP3 vesicles to the leading edge but also results in higher recycling rates. This is presumably the result of more VAMP3-containing recycling vesicles at the leading edge (Fig. 5c, d, Supplementary Movie 6).

VAMP3 is recycled via endosomal compartments, and based on the localization of WDFY2 to tubules, an attractive hypothesis was that lack of WDFY2 could lead to changes in VAMP3 recycling. To test this hypothesis, we transfected WT and WDFY2 (−/−) cells with GFP-VAMP3 and measured the distribution with different endocytic compartments. We observed that EEA1-positive early endosomes in WDFY2(−/−) cells showed reduced levels of VAMP3 (Fig. 5e), suggesting that either less VAMP3 arrives at the early endosomes or that recycling occurs faster. LAMP1-labeled vesicles also showed reduced VAMP3 levels

(Fig. 5f), indicating that the observed depletion of VAMP3 in early endosomes was not a result of a faster transport of cargo towards the degradative pathway.

Next, we asked if WDFY2 is required for recycling in general or if it is specifically required for sorting and recycling of VAMP3. Loss of WDFY2 did not affect the sorting of the canonical recycling cargo TFR, indicating that WDFY2 is not a general regulator of endocytic recycling (Supplementary Fig. 5d). We also tested if loss of WDFY2 affected endosomal tubulation. Using the WDFY2-derived 2xFYVE probe, which labels endosomal tubules but does not restore WDFY2 functionality, we measured the formation of tubules in wild-type (WT) and WDFY2(−/−) cells (Supplementary Fig. 5e, Supplementary Movies 7 and 8). Neither the formation nor the length of endosomal tubules was affected by the loss of WDFY2 (Supplementary Fig. 5f). However, we observed a slightly shortened lifetime of tubular structures in WDFY2(−/−) cells (Supplementary Fig. 5g), suggesting that these tubules are either not as stable or show an enhanced rate of vesicle formation.

**Loss of WDFY2 increases matrix metalloproteinase secretion.** In order to characterize the cellular function of WDFY2, we set out to find a cargo transported of the vesicles that accumulated at the leading edge of WDFY2(−/−) cells. VAMP3-positive vesicles have been shown to transport the membrane anchored matrix metalloproteinase MT1-MMP to the plasma membrane[4,27]. MT1-MMP has been reported to be transported via RAB4-dependent fast endocytic recycling, and RAB5A can redirect MT1-MMP to invadosomes in order to allow ECM degradation and cell invasion[28]. To investigate whether MT1-MMP could be a potential cargo for the WDFY2-positive tubules and VAMP3 containing vesicles, we stained cells expressing GFP-WDFY2 with antibodies against VAMP3 and MT1-MMP. We observed that both VAMP3 and MT1-MMP colocalized with GFP-WDFY2 on endosomes (Fig. 6a). Likewise, co-staining of GFP-WDFY2 with antibodies targeting RAB4 and MT1-MMP showed colocalization of WDFY2 with RAB4 and MT1-MMP (Fig. 6b).

Time-lapse imaging showed that GFP-MT1-MMP and mCherry-WDFY2 accumulated on the tubular domains of endosomes, with 58% of observed WDFY2 tubules containing MT1-MMP (Supplementary Fig. 6b, Supplementary Movie 9). Live-cell imaging showed that both GFP-MT1-MMP- and GFP-WDFY2-containing tubules are also positive for mCherry-RAB4, suggesting that these tubules give rise to recycling vesicles (Supplementary Fig. 6a, c, Supplementary Movies 10 and 11). Indeed, live-cell imaging of mCherry-VAMP3 and GFP-MT1-

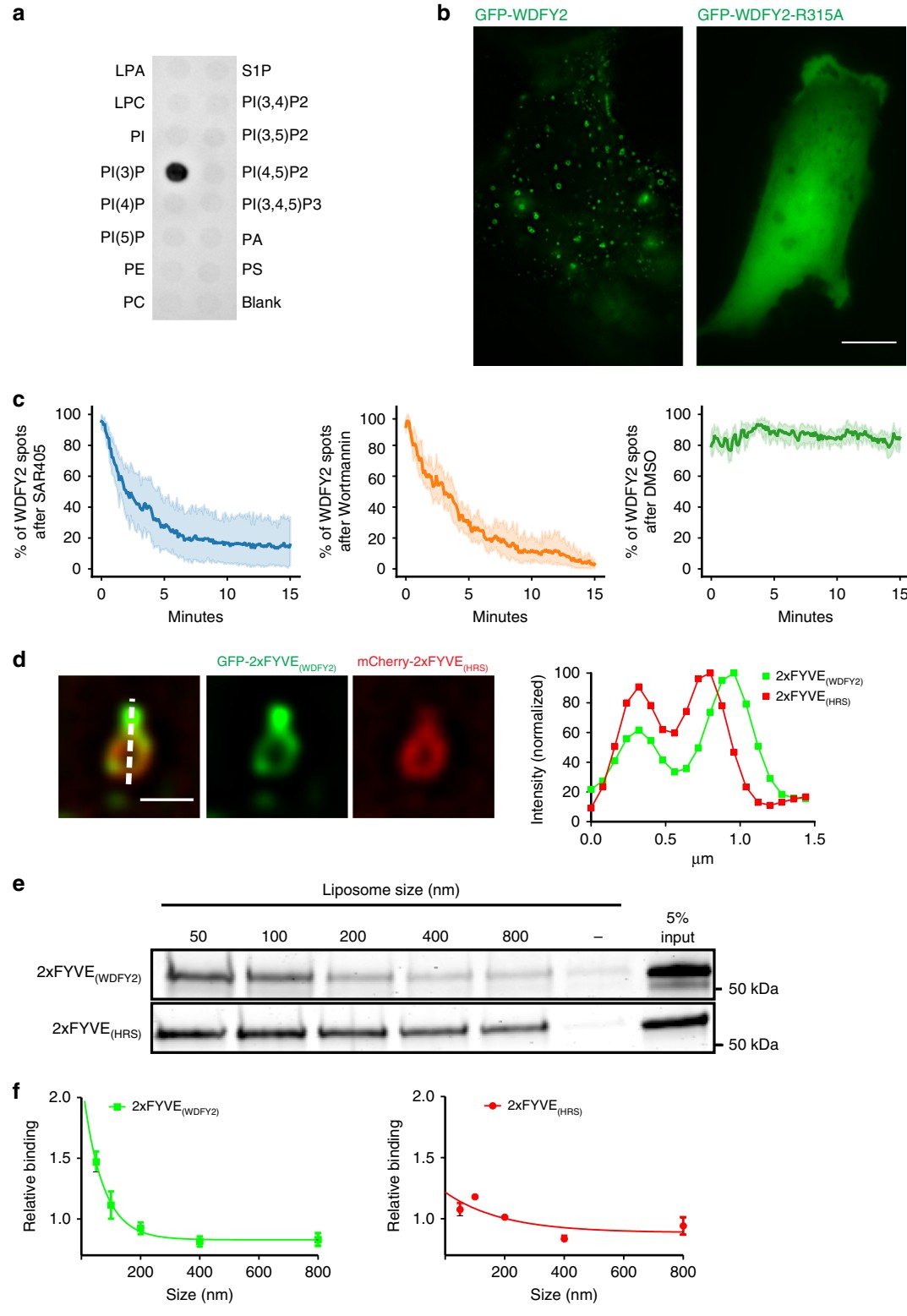

MMP showed VAMP3-labeled vesicles containing MT1-MMP (Supplementary Fig. 6d).

These observations raised the question if the secretion rate of MT1-MMP is affected by the loss of WDFY2. To investigate this, we performed TIRF live-cell microscopy of WT and WDFY2 (−/−) cells expressing pHluorin-MT1-MMP[29]. We observed that the rate of MT1-MMP exocytosis was strongly increased in the absence of WDFY2 (Fig. 6c, d, Supplementary Movie 12). Moreover, overexpression of WDFY2 led to a strong

accumulation of MT1-MMP in WDFY2-containing endosomes, suggesting that MT1-MMP is retained on endosomal compartments by WDFY2 (Fig. 6e, f). We did not observe enhanced protein levels of MT1-MMP upon overexpression of WDFY2 (Supplemental Fig. 6e, f), indicating that the observed accumulation is the consequence of a redistribution of MT1-MMP.

The exocytosis rate of MT1-MMP and the observed change in WDFY2(−/−) cells showed the same tendency as observed for VAMP3 exocytosis, leading us to hypothesize that VAMP3-

**Fig. 3** WDFY2 depends on PI3P for endosome localization. **a** Protein–lipid overlay assay using purified full-length WDFY2. WDFY2 binds with high selectivity to PtdIns3P. Shown is a representative plot from two experiments. **b** Deconvolved widefield image showing GFP-WDFY2 localization to endosomes. GFP-WDFY2-R315A, a mutation in the binding site for PtdIns3P, abolishes the localization to endosomes and the protein is cytosolic. Representative image from 10 cells per condition. Scale bar: 10 μm. **c** hTERT-RPE1 cells stably expressing GFP-WDFY2 was treated with SAR405, Wortmannin or DMSO (as a bleaching control) with a final concentration of 6 μM. Cells were imaged every 5 s for 15 min WDFY2 spots were quantified per time point (5 cells per condition). Shown are mean ± 95% CI. **d** Distribution of GFP-2xFYVE$_{WDFY2}$ and mCherry-2xFYVE$_{HRS}$ on a tubulating endosome. A WDFY2-derived 2xFYVE probe shows a preference for tubulating membranes. The graph shows the normalized fluorescence intensity of GFP-2xFYVE$_{WDFY2}$ and mCherry-2xFYVE$_{HRS}$ along the indicated line. Representative image from 50 observed endosomes. Scale bar: 1 μm. **e** Coomassie Brilliant Blue stained gel showing top fractions from liposome flotation assays using His-MPB-fused of 2xFYVE$_{WDFY2}$ and 2xFYVE$_{HRS}$ with differently sized liposomes. Shown is one representative gel from three independent experiments. **f** Plots showing the relative binding of 2xFYVE$_{WDFY2}$ and 2xFYVE$_{HRS}$ to differently sized liposomes. Data shown are derived from three experiments. Shown are mean ± S.E.M. Source data are provided as a Source Data file

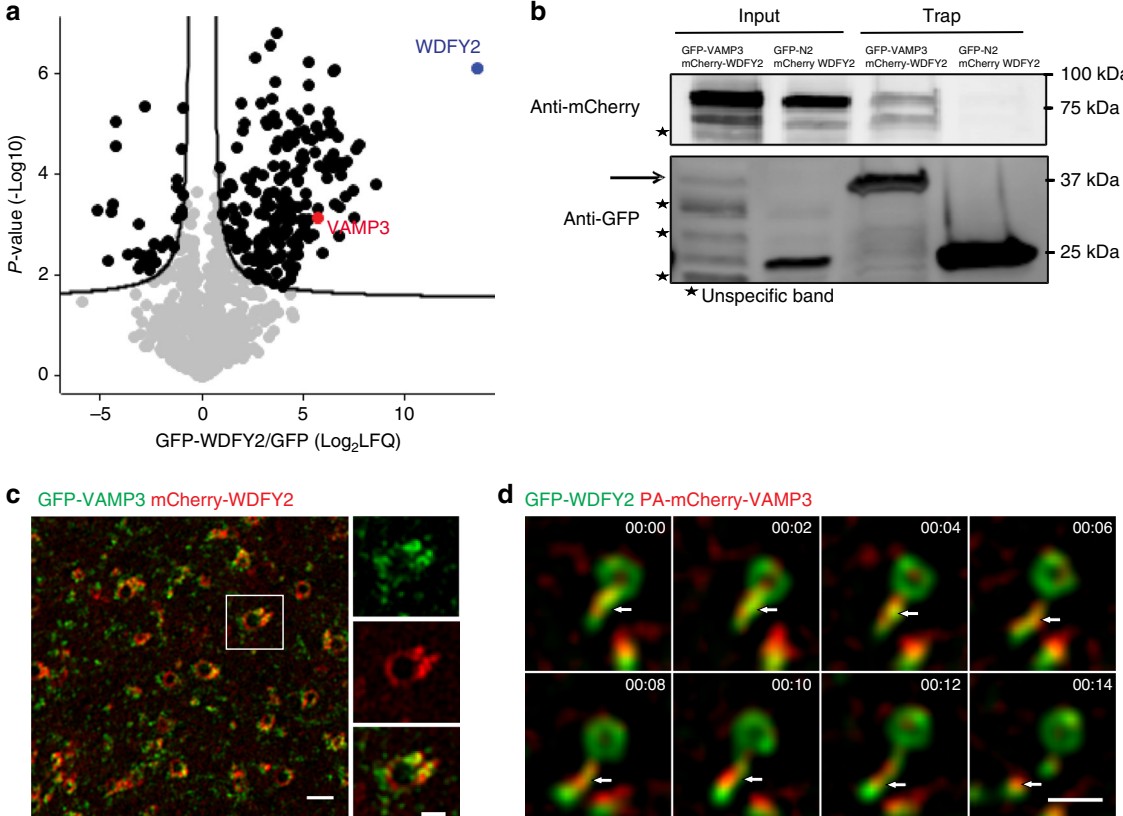

**Fig. 4** WDFY2 interacts and colocalizes with VAMP3. **a** Volcano plot identifying enriched proteins isolated from cells expressing GFP-WDFY2 by GFP-trap affinity purification of cell lysates in comparison to GFP-expressing control cells. Label-free quantification (LFQ) ratios of GFP-WDFY2/GFP pulldowns were plotted against the p-value, with the permutation-based FDR threshold (<0.01) indicated by the black lines. **b** GFP affinity purification of GFP-VAMP3 and mCherry-WDFY2. Affinity isolation of GFP-VAMP3 coprecipitates mCherry-WDFY2, whereas isolation of GFP alone does not coprecipitate mCherry-WDFY2. One representative blot from three experiments is shown. **c** Representative SIM image showing GFP-VAMP3 and mCherry-WDFY2 localization to endosomes and endosomal tubules. Scale bar overview image: 1 μm, scale bar of inset image: 0.5 μm. Representative image from 13 observed cells. **d** Sequential images showing GFP-WDFY2 and PA-mCherry-VAMP3 localization to endosomes and endosomal tubule. The tubule pinches off and remains positive for VAMP3. Shown are frames from a time-lapse movie with images acquired every second. Representative image from 20 cells. Scale bar: 1 μm. Source data are provided as a Source Data file

driven exocytosis of MT1-MMP might be controlled by WDFY2. In order to test this, we measured whether depletion of VAMP3 would suppress the enhanced secretion of MT1-MMP in cells lacking WDFY2. Indeed, knockdown of VAMP3 reduced secretion of pHluorin-MT1-MMP (Fig. 6g, Supplementary Fig. 6g). We therefore conclude that WDFY2 controls secretion of MT1-MMP via regulation of VAMP3.

Based on our observation that more MT1-MMP was secreted from cells lacking WDFY2, we tested if WDFY2(−/−) cells were able to degrade more ECM. To this end, we assayed degradation of gelatin labeled with the fluorescent dye Oregon green. WT and WDFY2(−/−) cells were seeded on coverslips coated with a layer of fluorescent gelatin[30], fixed 6 h after seeding, and stained with Phalloidin to visualize cells. Gelatin degradation was visible as black areas in the gelatin layer. We observed that WDFY2(−/−) cells degraded visibly more gelatin in comparison to WT cells (Fig. 7a, c). Incubation of WDFY2(−/−) cells with the MMP inhibitor GM6001 resulted in a complete lack of degradation, indicating that the observed effects were indeed based on secretion and function of matrix-metalloproteinases (Fig. 7a, c).

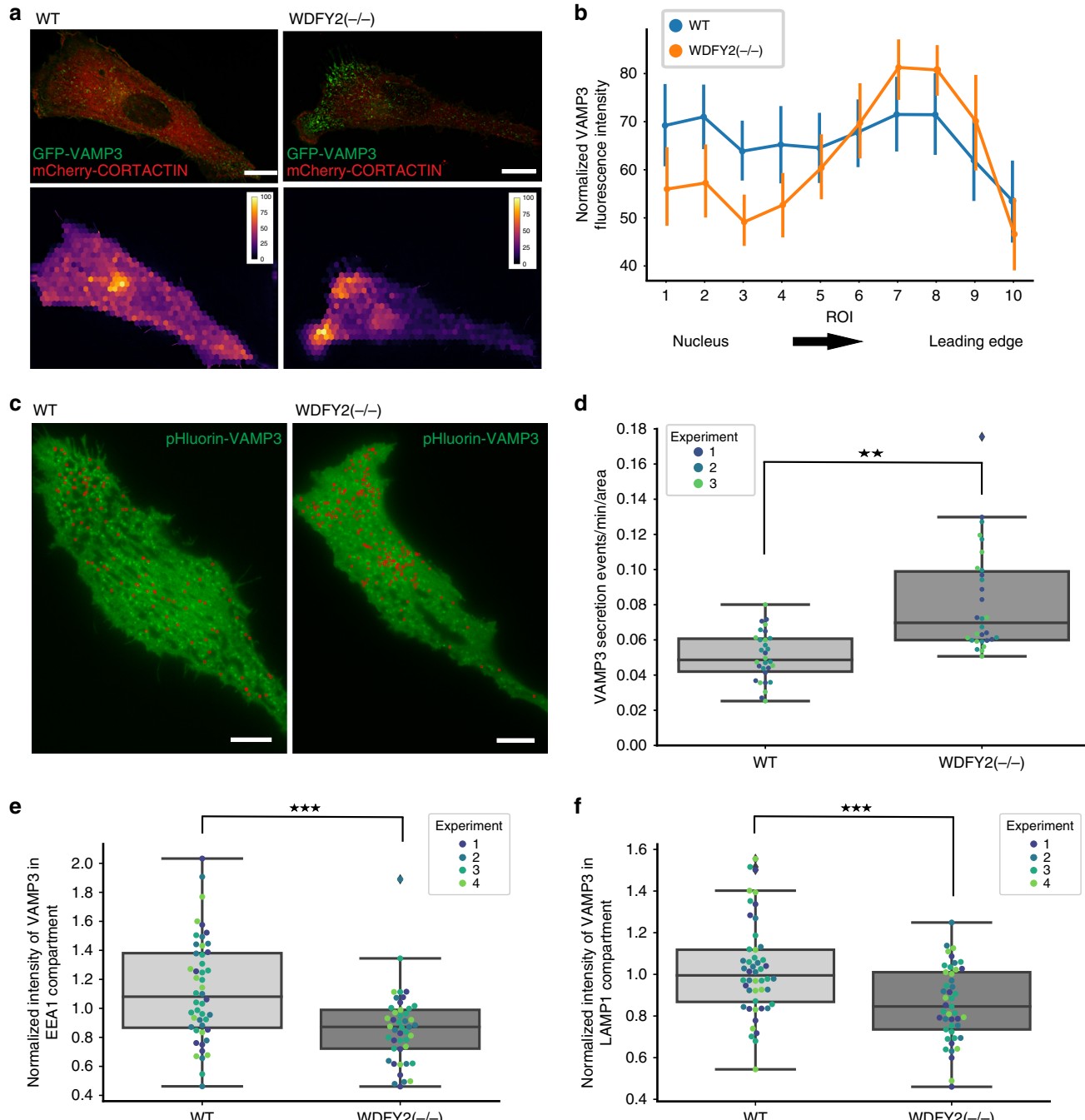

Likewise, siRNA-based depletion of MT1-MMP in WDFY2(−/−) cells completely blocked gelatin degradation (Fig. 7b, d Supplementary Fig. 7e), demonstrating that the observed enhanced gelatin degradation in WDFY2(−/−) cells was specifically mediated by MT1-MMP.

To test if this effect is mediated by VAMP3-dependent secretion of MT1-MMP, we depleted VAMP3 in WDFY2(−/−) cells and measured whether these cells were still able to degrade gelatin. Quantification of gelatin degradation showed that depletion of VAMP3 in WDFY2(−/−) cells suppressed the degradation of gelatin (Fig. 7e), in line with our finding that the enhanced secretion of MT1-MMP was VAMP3-dependent. This indicates that loss of WDFY2 not only enhances exocytosis of pHluorin-MT1-MMP, but also affects trafficking of endogenous MT1-MMP and thereby its ability to degrade ECM.

**Loss of WDFY2 increases 3D invasion of noncancerous cells.** Cancer cells often show elevated levels of MMPs and increased secretion, which allows them to invade neighboring tissues and establish metastases[5]. The elevated MT1-MMP secretion and degradation of ECM in WDFY2(−/−) cells led us to ask if this allowed cells lacking WDFY2 to invade into 3D matrices. To this end, we performed 3D inverted invasion assays in Matrigel[31]. WT and WDFY2(−/−) cells were seeded onto the bottom of a Transwell filter and allowed to migrate through the filter and into a Matrigel plug. Cell invasion was analyzed by confocal microscopy. We observed that WDFY2(−/−) cells were able to invade further into Matrigel in comparison to wild-type control cells (Fig. 8a–c). This observation was supported by siRNA-based depletion of WDFY2, which showed the same phenotype as a full knockout (Supplementary Fig. 7a, d). Importantly, the effect on WDFY2 depletion could be rescued by stable expression of a

**Fig. 5** WDFY2 controls intracellular VAMP3 distribution. **a** Deconvolved widefield images of WT and WDFY2(−/−) cells transiently expressing GFP-VAMP3 and mCherry-CORTACTIN. Scale bar: 10 µm (representative image of 30 cells per condition. Hexagonal superpixel image showing VAMP3 distribution in the cells shown in the upper panel. Mean intensities of hexagonal ROIs were extracted and the ROI filled with the corresponding mean value, thereby generating superpixels. **b** Distribution of VAMP3 from the nucleus to the leading edge. Cells were transfected with VAMP3 and stained for CORTACTIN to identify leading edges. A line ROI from the nucleus to the leading edge was drawn, and evenly spaced ROIs were automatically generated. Mean intensity of VAMP3 in each ROI was extracted, normalized for each cell, and then plotted. 30 cells for each condition, shown are mean and 95% CI. **c** TIRF micrograph of WT and WDFY2(−/−) cells transfected with VAMP3-pHluorin. Individual secretion events (summed over a 2-min interval) are indicated with a red circle. Representative image from 30 cells per condition. Scale bar: 10 µm. **d** Quantification of VAMP3-pHluorin secretion in WT and WDFY2(−/−) cells, as shown in (**c**). Shown are events from three experiments, 10 cells per experiment per condition. Secretion events per minute are shown normalized to cell area. Student's unpaired $t$ test, $p = 0.00205$. Shown are the median, quartiles (boxes), and 1.5 times the interquartile range (whiskers). *$p < 0.05$, **$p < 0.01$, ***$p < 0.001$, n.s. not statistically significant. **e** Quantification of GFP-VAMP3 intensity in EEA1-positive endosomes in WT and WDFY2(−/−) cells. Plotted is the mean intensity within EEA1-positive endosomes per cell from four independent experiments (10, 12, 15, 10 cells (WT), 10, 10, 15, 10 cells (WDFY2(−/−)). Student's unpaired $t$ test, $p = 0.00015$. Shown are the median, quartiles (boxes), and 1.5 times the interquartile range (whiskers). *$p < 0.05$, **$p < 0.01$, ***$p < 0.001$, n.s. not statistically significant. **f** Quantification of GFP-VAMP3 intensity in LAMP1-positive vesicles in RPE1 (WT) and WDFY2(−/−) cells. Plotted is the mean intensity within LAMP1-positive vesicles per cell from four experiments (10, 12, 15, 10 cells (WT), 10, 10, 15, 10 cells (WDFY2(−/−)). Student's unpaired $t$ test, $p = 0.00083$. Shown are the median, quartiles (boxes), and 1.5 times the interquartile range (whiskers). *$p < 0.05$, **$p < 0.01$, ***$p < 0.001$, n.s. not statistically significant. Source data are provided as a Source Data file

siRNA-resistant WDFY2 construct, confirming the involvement of WDFY2 in cell invasion (Supplementary Fig. 7a, c).

To validate that the observed invasion is mediated by enhanced secretion of MT1-MMP, we performed inverted invasion assays into collagen matrices, since invasion into collagen is strictly dependent on MT1-MMP[32]. Like with Matrigel as matrix material, we observed that cells deleted for WDFY2 showed increased invasion into collagen (Fig. 8d). Taken together, this suggests that WDFY2 is critical to control MT1-MMP-mediated cell invasion in 3D.

**WDFY2 controls the invasivity of cancer cells**. WDFY2 is frequently lost in metastatic tumors, but it is not known if there is a direct link between loss of WDFY2 and the metastatic potential of tumor cells. The observation that lack of WDFY2 enhanced MT1-MMP secretion led us to speculate that this increased secretion could contribute to the metastatic potential of cancer cells. To test this hypothesis, we used two invasive cancer cell lines, MDA-MB231 breast cancer cells and PC3 prostate cancer cells, as model systems.

In MDA-MB231 cells, like in hTERT-RPE1 cells, WDFY2 localized to tubular regions of endosomes (Fig. 9a). We also observed colocalization of mCherry-WDFY2 with MT1-MMP on EEA1-positive endosomes (Fig. 9b). MDA-MB231 cells are a well-established model system for 3D invasion, and we therefore asked if WDFY2 controls cell invasion in this cell type. We depleted WDFY2 using siRNA (Supplementary Fig. 8a) and tested if MDA-MB231 cells lacking WDFY2 showed increased invasion in Matrigel. As observed for RPE1 cells, depletion of WDFY2 led to increased invasion in Matrigel, which could be rescued by expression of a siRNA-resistant WDFY2 allele (Fig. 9c, d).

PC3 cells have been shown to secrete high levels of matrix metalloproteinases and show an invasive phenotype[33,34]. Because PC3 cells express low levels of WDFY2[35], we asked if elevated expression of WDFY2 could reduce the invasive phenotype of these cells. To this end, we generated stable PC3 cell lines expressing GFP-tagged WDFY2. Fluorescence microscopy showed that also in these cells, WDFY2 localizes to endosomal tubules (Fig. 9e). Using inverted invasion assays, we assessed the invasive potential of the parental cell line and cells overexpressing WDFY2. Whereas parental cells were able to invade into Matrigel, cells overexpressing WDFY2 showed strongly reduced invasion. These result support a role of WDFY2 in negative control of cancer cell invasion (Fig. 9f, g).

## Discussion

In this study, we provide evidence that tumor suppressors can act by restricting endocytic recycling of MMPs and thereby preventing cell invasion. We show that the endosomal protein WDFY2 regulates endocytic recycling to the plasma membrane of MT1-MMP via a VAMP3-dependent mechanism, and that knockout of WDFY2 causes increased degradation of ECM accompanied by increased cell invasion.

We found that WDFY2 localized to tubules emanating from EEA1-positive early endosomes which were positive for the retromer complex, the actin bundling protein CORONIN1B, and the small GTPase RAB4. We observed that vesicles were formed from these tubules which carried recycling cargoes such as VAMP3 and MT1-MMP. This suggests that WDFY2 could localize to structures that regulate cargo sorting for recycling.

Localization of WDFY2 was dependent on a functional FYVE domain, and the full-length protein binds with high specificity to the PI PtdIns3P. Consequently, inhibition of PtdIns3P production rapidly displaced WDFY2 from endosomal tubules. Surprisingly, when probing PtdIns3P localization with the canonical HRS-derived 2xFYVE probe, we observed only weak labeling of tubular structures on endosomes, whereas a WDFY2-derived 2xFYVE probe showed a clear preference for endosomal tubules. This argues that the WDFY2 FYVE domain has different binding properties in comparison to the HRS FYVE domain; while both can bind PtdIns3P, they detect distinct pools. This observation suggests that localization of FYVE domain containing proteins does not only depend on the pure presence of PtdIns3P, but that FYVE domains can recognize and preferentially bind to PtdIns3P in a specific membrane context. Indeed, we found that this distinct binding is caused by sensing of membrane curvature by the WDFY2 FYVE domain. Mechanistically, this is likely caused by a short insertion in the turret loop of the WDFY2 FYVE domain[6]. FYVE domains bind to PtdIns3P-containing membranes by inserting this hydrophobic loop into the membrane[36]. A larger turret loop could sterically hinder insertion into flat membranes and could drive the WDFY2 domain to highly curved endosomal tubules. Our findings also demonstrate the limitations of the current generation of PtdIns3P sensors, and our study provides a sensor able to detect PtdIns3P on tubular membrane structures.

We identified the v-SNARE VAMP3 as an interactor of WDFY2 and established that cells deleted for WDFY2 show an accumulation of VAMP3-labeled vesicles in the cell periphery and higher recycling rates of VAMP3-positive vesicles. A cargo of VAMP3 vesicles, MT1-MMP[27], showed the same behavior. This

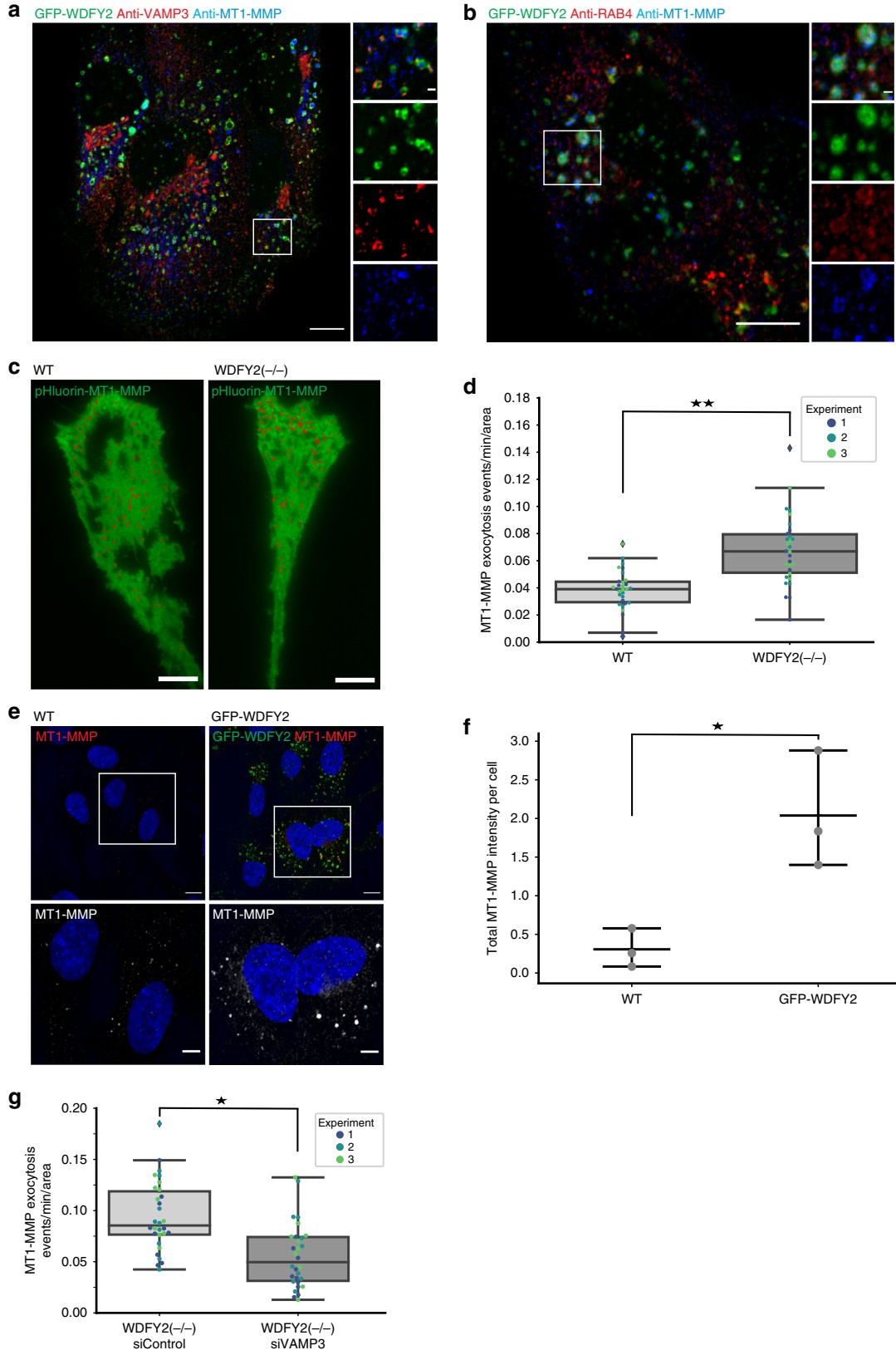

raises the question how WDFY2 could influence VAMP3-dependent trafficking.

Sorting and retrieval of endocytosed cargo is one of the fundamental functions of the endocytic system. Tubular membrane structures are thought to play a major role in this process, but it is largely unknown how their cargo is selected[12]. On early endosomes, several sorting pathways have been described which are involved in trafficking of different cargos. Some cargos, such as TFR, are sorted into short-lived tubules, and it has been shown that TFR can be sorted into these structures by fast diffusion[12]. In contrast to this, slow-diffusing cargos like β2-adrenergic receptors are sorted into stable actin-dependent membrane tubules[12]. These

**Fig. 6** WDFY2 controls MT1-MMP trafficking. **a** Confocal images showing localization of MT1-MMP and VAMP3 to GFP-WDFY2-positive endosome. Representative image from 10 cells. Scale bar: 10 μm, inset: 1 μm. **b** Confocal images showing localization of MT1-MMP and RAB4 to GFP-WDFY2-positive endosome. Representative image from 10 cells. Scale bar: 10 μm, inset: 1 μm. **c** TIRF micrograph of WT and WDFY2(−/−) cells transfected with pHluorin-MT1-MMP. Individual exocytic events (summed over two minutes) are indicated with a red circle. Representative image from 30 observed cells per condition. Scale bar: 10 μm, **d** Quantification of pHluorin-MT1-MMP exocytosis in WT and WDFY2(−/−) cells. Shown are events from three experiments, 10 cells per experiments per condition. Exocytosis events per minute are shown normalized to cell area. Student's unpaired t test, $p = 0.0026$. Shown are the median, quartiles (boxes), and 1.5 times the interquartile range (whiskers). *$p < 0.05$, **$p < 0.01$, ***$p < 0.001$, n.s. not statistically significant. **e** Confocal micrographs of RPE1 (WT) and RPE1 cells stably expressing GFP-WDFY2. Cells overexpressing WDFY2 accumulates MT1-MMP in endosome positive for WDFY2. Scale bar: 10 μm, insets: 5 μm. **f** Quantification of MT1-MMP spots based on high-content microscopy show a significant increase in total intensity of MT1-MMP in hTERT-RPE1 GFP-WDFY2 cells when compared to hTERT-RPE1 parental cells. Shown are quantifications from three independent experiments, cells in total: hTERT-RPE1: 3570, hTERT-RPE1 GFP-WDFY2: 5250. Plotted are individual experiments and the mean ± 95% CI. $p = 0.02007$. *$p < 0.05$, **$p < 0.01$, ***$p < 0.001$, n.s. not statistically significant. **g** Quantification of pHluorin-MT1-MMP exocytosis in WDFY2(−/−) cells treated with siControl or siVAMP3. Shown are events from three experiments, ten cells per experiments per condition. Exocytosis events per minute are shown normalized to cell area. Student's unpaired t-test, $p = 0.0149$. *$p < 0.05$, **$p < 0.01$, ***$p < 0.001$, n.s. not statistically significant. Source data are provided as a Source Data file

tubules are relatively long-lived and can form vesicles which are recycled back to the plasma membrane. One of their key characteristics is that they recruit actin and actin-binding proteins such as CORONIN1B and WASH which stabilize the tubules and allow them to form stable sorting platforms[12]. The tubular structures labeled by WDFY2 were positive for CORONIN1B and WASH and sensitive to actin disruption by LatrunculinB, indicating that they could represent actin-stabilized tubules. WDFY2 and VAMP3 colocalized at these structures, suggesting that WDFY2 interacts with VAMP3 at these actin-stabilized tubules.

Loss of WDFY2 led to enhanced secretion of VAMP3-positive vesicles and their cargo, MT1-MMP. How could the loss of WDFY2 lead to these changes? We propose the following model (Fig. 10a): Based on the characteristics of WDFY2-positive tubules, it is an attractive hypothesis that WDFY2 normally controls sorting of VAMP3 into stable, actin-dependent recycling tubules. Interaction of WDFY2 with VAMP3 could slow down diffusion and thereby restrict the number of v-SNARE molecules on recycling vesicles. A loss of WDFY2 would abolish this control and allow VAMP3 to be sorted into bulk recycling tubules, leading to increased recycling.

Consistent with this, WDFY2(−/−) cells showed less VAMP3 in both EEA1 and LAMP1 vesicles, supporting the notion that loss of WDFY2 leads to altered sorting or recycling of VAMP3. The increased secretion of MT1-MMP in the absence of WDFY2 is likely a direct consequence of altered VAMP3 sorting by providing more recycling vesicles for fusion with the plasma membrane (Fig. 9b). It is not clear how the number of v-SNARE molecules could influence the rate of membrane fusion, but one explanation could be that more newly formed vesicles gain VAMP3 and thereby a secretory identity, potentially by recruiting PI4K2A and generation of PtdIns4P[22].

Several lines of evidence suggest that WDFY2 can act as a tumor suppressor. A screen of the cBioportal cancer genome database shows that WDFY2 is frequently (in up to 14% of cases) lost in cancers (Fig. 10b)[37,38]. An earlier study reported a *CDKN2D-WDFY2* fusion gene, which occurs frequently in high-grade serous ovarian cancer (in 20% of all HG-SC tumors)[39]. The fusion leads to expression of a truncated WDFY2 protein[39]. It is likely that this fusion protein would be unable to control VAMP3 trafficking, as part of the first WD repeat is missing and the truncated protein would not form a functional β-propeller. The loss of WDFY2 in cancer cells could enable them to migrate through the ECM and provide a higher metastatic potential, which correlates well with the finding that WDFY2 is frequently lost in cancers. In line with this, we find that depletion of WDFY2 in MDA-MB231 cells enhances 3D invasion, whereas over-expression of WDFY2 in invasive PC3 cells—which have been

shown to have high levels of MMP activity—reduces their invasive potential.

We conclude that WDFY2 normally acts as a traffic gatekeeper which limits cell invasion by restraining VAMP3-dependent recycling of MT1-MMP from endosomes to the plasma membrane. A loss of this control mechanism increases MT1-MMP secretion, ECM degradation and cell invasivity and is likely to increase the metastatic potential of cancer cells. In future studies it will be interesting to test this in preclinical models.

## Methods

**Antibodies**. The following antibodies were used: Human anti-EEA1 provided by Ban-Hock Toh (Monash University, Immunofluorescence 1:160,000), Rabbit anti-APPL1 D83H4 from cell signaling (3858S, Immunofluorescence 1:100), Rabbit anti-RAB7 was from Cell Signaling (9367, Immunofluorescence 1:200), Rabbit anti-RAB11 was from Zymed Laboratories (71-5300, Immunofluorescence 1:100), Mouse anti-RAB5 was provided by C. Bucci (University of Salento, Immunofluorescence 1:2500), Rabbit anti-RAB4 was from Fisher Scientific (PA3–912, Immunofluorescence 1:200), Mouse anti-GFP was from Roche (11814 460001, Immunofluorescence 1:400, western blot 1:1000), RFP-booster ATTO-594 was from Chromotek (rba594, Immunofluorescence 1:500), Rabbit antibody against HRS have been described previously[40] (Immunofluorescence 1:100). Rabbit anti-LAMP1 was from Sigma-Aldrich (L1418, Immunofluorescence 1:200), Rabbit anti-VAMP3 was from Synaptic Systems (104,203, Immunofluorescence 1:200, Western blot 1:1000), Mouse anti-MT1-MMP was from Merck Life science (MAB3328, Immunofluorescence 1:800, western blot 1:1000), Rhodamine Phalloidin (Thermo Fisher, R415), Sheep anti-TGN46 was from AbD Serotec (AHP500G, Immunofluorescence 1:100), Mouse anti-γ-TUBULIN (T6557, western blot 1:10,000) and mouse anti-α-TUBULIN (T5168, western blot 1:20,000) were from Sigma-Aldrich, Hoechst 33342 (H3570) was from Invitrogen Molecular Probes, Goat anti-VPS35 (ab10099, Immunofluorescence: 1:100), Rabbit anti-VPS26 (ab23892, Immunofluorescence: 1:100) and Rabbit anti-β-TUBULIN (ab6046, western blot 1:1000) were from Abcam. Goat anti-mCherry was from Acris Antibodies (AB0040-200, Immunofluorescence: 1:100, western blot 1:1000). HRP-conjugated anti-GST antibody was from GE Healthcare (RPN1236, western blot 1:5000). Secondary antibodies used for IF and western blotting were obtained from Jackson ImmunoResearch Laboratories and LI-COR Bioscience GmbH.

**Plasmids**. pmCherry-Rab11a was a gift from Jim Norman[41], pCDNA-pHlourin_MT-MMP was gift from Philippe Chavrier[42], for some experiments, the pHluorin tag was exchanged with eGFP, Fam21-GFP (pEGFP-N1–3) was a gift from Dr. Matthew Seaman[14]. The NLAP cassette used for endogenous tagging was a gift from Anthony Hyman[43].

The following plasmids were obtained from Addgene: pmCherry-RAB4 (55125) and pmCherry-WASH1-C-18 (55162) were a gift from Michael Davidson, pEGFP-VAMP3 (42310) was a gift from Thierry Galli[44]. pmCherry-CORTACTIN (27676) and CORONIN1B-mCherry (27694) were a gift from Christien Merrifield[45]. pX458 (48138) was a gift from Feng Zhang[24]. AAV-2A-Neo V2 (80033) was a gift from Hiroyuki Konishi[46]. To create pVAMP3-pHlourin, VAMP3 and supereclipticpHlourin were polymerase chain reaction (PCR)-amplified and cloned into a pEGFP-N1-derived backbone.

All other plasmids were generated using standard cloning procedures and are listed in Supplementary Table 1. Detailed cloning procedures can be obtained from the authors.

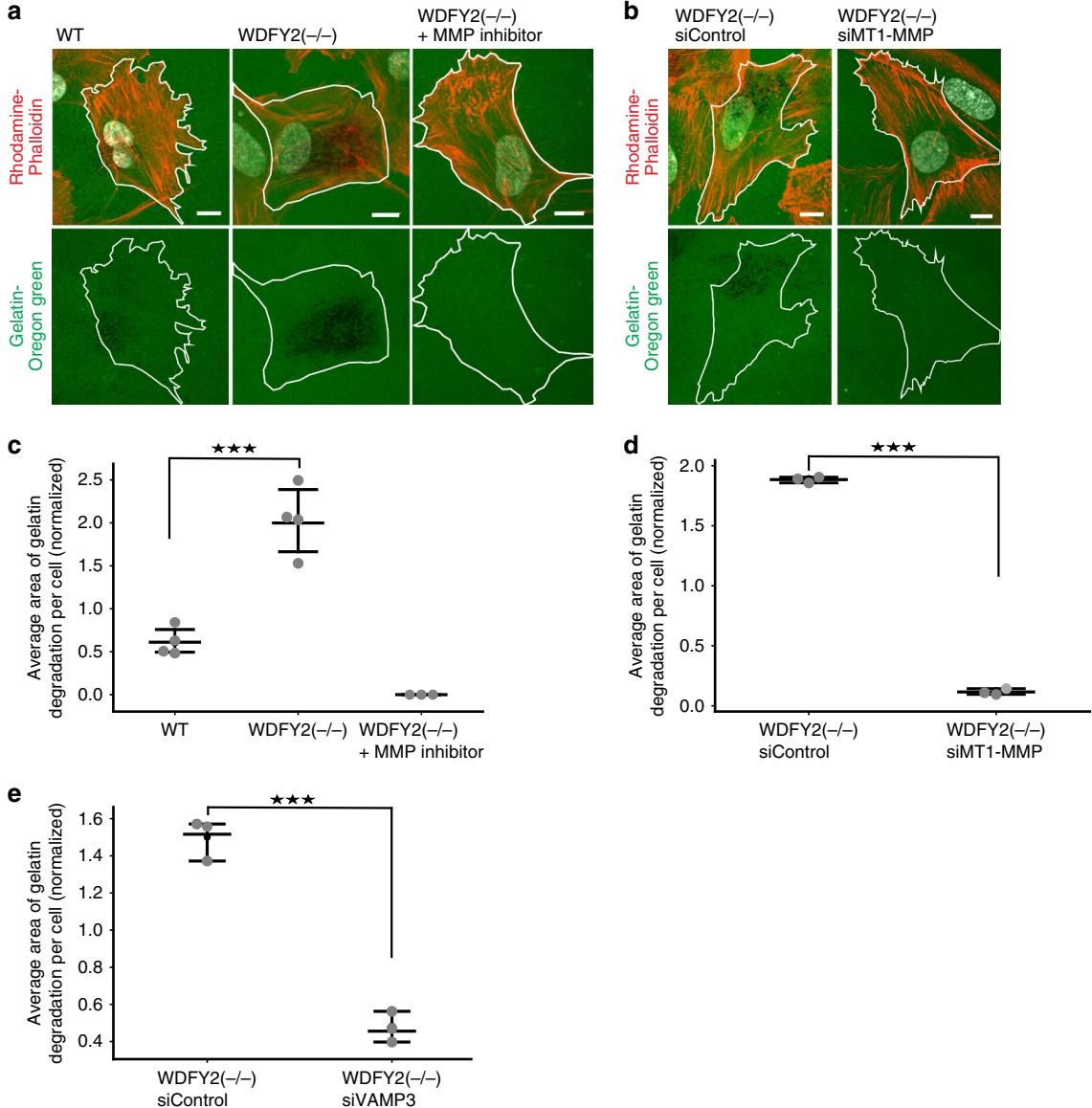

**Fig. 7** WDFY2 controls extracellular matrix degradation. **a** Confocal micrographs showing degradation of a fluorescent gelatin layer, indicated by dark areas, by WT, WFDY2(-/-) and MT1-MMP-inhibitor treated WFDY2($-/-$) cells. Scale bar: 5 μm. **b** Confocal micrographs showing degradation of a fluorescent gelatin layer, indicated by dark areas, in WDFY2($-/-$) cells treated with control siRNA and MT1-MMP siRNA. Scale bar: 5 μm. **c** Quantification of gelatin degradation shown in (**a**). The graph shows average area of gelatin degradation per cell. $n = 4$ experiments, representing in total 815 cells (WT), 741 cells (WDFY2($-/-$)) and 221 cells (WDFY2($-/-$) + MMP inhibitor). Data points indicate the mean degradation per experiment, also shown is the mean ± 95% CI. Student's unpaired $t$ test, $p = 0.00057$. *$p < 0.05$, **$p < 0.01$, ***$p < 0.001$, n.s. not statistically significant. **d** Quantification of gelatin degradation shown in (**b**). The graph shows average area of gelatin degradation per cell $n = 3$ experiments, representing 543 cells (WDFY2($-/-$) siControl) and 455 cells (WDFY2($-/-$) MT1-MMP siRNA) in total. Data points indicate the mean degradation per experiment, also shown is the mean ± 95% CI. Student's unpaired $t$ test, $p = 1.028e-07$. *$p < 0.05$, **$p < 0.01$, ***$p < 0.001$, n.s. not statistically significant. **e** Quantification of gelatin degradation in WDFY2($-/-$) cells treated with control siRNA and VAMP3 siRNA. The graph shows average area of gelatin degradation per cell $n = 3$ experiments, representing 618 cells (WDFY2($-/-$) siControl) and 610 cells (WDFY2($-/-$) VAMP3 siRNA) in total. Data points indicate the mean degradation per experiment, also shown is the mean ± 95% CI. $p = 0.0002164$. Student's unpaired $t$ test, *$p < 0.05$, **$p < 0.01$, ***$p < 0.001$, n.s. not statistically significant. Source data are provided as a Source Data file

**Cell culture**. hTERT-RPE1 cell cultures were maintained in DMEM-F12 medium (Gibco) with 10% fetal bovine serum (FBS), 5 U ml$^{-1}$ penicillin and 50 μg ml$^{-1}$ streptomycin at 37 °C and 5% $CO_2$. PC3 cell cultures were maintained in DMEM/F12 medium (Gibco) with 7% FBS, 5 U ml$^{-1}$ penicillin and 50 μg ml$^{-1}$ streptomycin at 37 °C and 5% $CO_2$. MDA-MB231 cell cultures were penicillin and 50 μg ml$^{-1}$ streptomycin at 37 °C and 5% $CO_2$.

**Cell lines**. Experiments were performed in hTERT-RPE1 (CRL-4000), PC3 (CRL-1435), or MDA-MB231 (HTB-26) cells. Cells were purchased from ATCC. All cells were verified to be free of mycoplasma contamination and regularly tested after manipulation. Their identity was verified using Powerplex16 assays. Stable hTERT-RPE1, PC3, or MDA-MB231 cell lines were lentivirus generated pools. For all cell lines a PGK promoter was used. Third generation lentivirus was generated as follows[47]: GFP or mCherry fusions were generated as Gateway ENTRY plasmids using standard molecular biology techniques. From these vectors, lentiviral transfer vectors were generated by recombination into Lentiviral "Destination" vectors derived from pCDH-EF1α-MCS-IRES-PURO (SystemBioSciences) using a Gateway LR reaction. VSV-G pseudotyped lentiviral particles were packaged using a

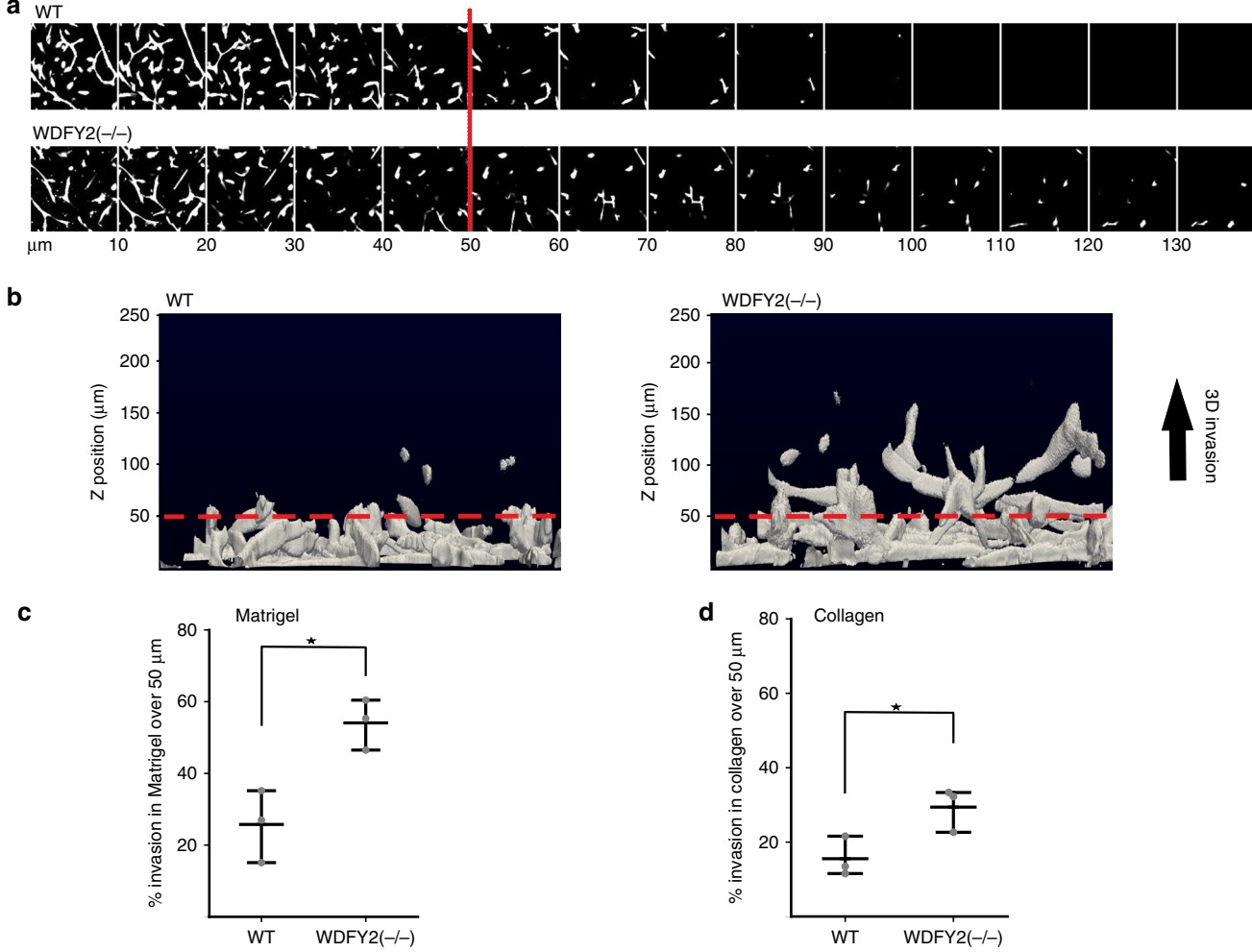

**Fig. 8** WDFY2 controls invasive cell migration. **a** Optical sections ($\Delta z = 10\,\mu m$) of WT and WDFY2(−/−) cells stained with Calcein-AM invading into fibronectin-supplemented Matrigel™. The red line indicates the z-axis threshold (50 μm) defining invading cells. **b** Orthogonal view of a 3D reconstruction of WT and WDFY2(−/−) invading into fibronectin-supplemented Matrigel™. The red line indicates the z-axis threshold (50 μm) defining invading cells. **c** Quantification of invasion of WT and WDFY2(−/−) cells into fibronectin-supplemented Matrigel™. Plotted points indicate the mean invasion per experiments (15 z-stacks per experiment, n = 3 experiments), also shown is the mean ± 95% CI. Student's unpaired t test, p = 0.0162. *p < 0.05, **p < 0.01, ***p < 0.001, n.s. not statistically significant. **d** Quantification of invasion of WT and WDFY2(−/−) cells into fibronectin-supplemented acid extracted type 1 Collagen. Plotted points indicate the mean invasion per experiments (15 z-stacks per experiment, n = 3 experiments), shown is the mean ± 95% CI. Student's unpaired t test, p = 0.0389. *p < 0.05, **p < 0.01, ***p < 0.001, n.s. not statistically significant. Source data are provided as a Source Data file

third generation packaging system (Addgene plasmids numbers 12251, 12253, and 12259)[48]. Cells were then transduced with virus and stable expressing populations were generated by antibiotic selection. Used cell lines are listed in Supplementary Table 2.

**Immunostaining**. Cells grown on coverslips were fixed with 3% formaldehyde (Polyscience) for 15 min in room temperature and permeabilized with 0.05% Saponin (Sigma-Aldrich) in PBS. Cells were then stained with the indicated primary antibodies for 1 h, washed in PBS/Saponin, stained 1 h with fluorescently labeled secondary antibodies and washed with PBS/Saponin. Cells were mounted using Mowiol (Sigma-Aldrich). Cells stained for RAB7 were pre-permeabilized in 0.05 % Saponin in PEM buffer (80 mM K-Pipes, 5 mM EGTA, 1 mM MgCl₂ (pH 6.8)) for 5 min in room-temperature, before fixation in 3% formaldehyde.

**Transient transfection**. hTERT-RPE1 cells were transfected with Fugene6 using 3 μl reagent per 1 μg of DNA. Cells for live-cell imaging were transfected in MatTek dishes (Inter Instruments) dishes and cells for fixation were transfected in 24-well plates with coverslips.

**siRNA-mediated protein knockdown**. All siRNA used was obtained from Ambion. Cells were transfected using Lipofectamine RNAiMAX transfection reagent (Life Technologies) following the manufacturers protocol. Totally, 50 nM siRNA targeting WDFY2 (sense: GCAUGUCUUUUAACCCGGA) (s41881) was

used. For depletion of MT1-MMP, Silencer™ select siRNA (sense: GCAACAUA AUGAAAUCACU)(s8879) was used. For depletion of VAMP3, Silencer™ select siRNA (sense: CGGGAUUACUGUUCUGGUU)(s17856) was used. Silencer™ select negative control No. 2 siRNA (AM4637) was used as a control. Transfection was performed using the RNAiMax "reverse transfection" protocol.

**CRISPR/Cas9-mediated deletion of WDFY2**. hTERT-RPE1 cells deleted for WDFY2 were generated using CRISPR/Cas9. Guide RNAs were designed using Benchling software (www.benchling.com). For deletion of *WDFY2*, a guide RNA binding in directly in front of the start codon in exon1 and one binding after exon1 was chosen (gRNA1: 5′-TGGATCTCCGCCGCCATCGG-3′; gRNA2: 5′-ACTACTGCCATTCGGCCGCG-3′). This strategy resulted in deletion of the first 46 amino acids of WDFY2 and deleted also the splice donor of the intron linking exon 1 and 2 (Supplementary Fig. 5). We reasoned that this deletion should not result in a functional cDNA due to deletion of the splice site.

pX458-derived plasmids encoding both Cas9-2A-GFP and the respective gRNA were transfected using Fugene6[24]. Forty-eight-hour post-transfection, GFP-positive cells were sorted and seeded out in several dilutions to obtain single colonies, which were picked and characterized. Clones lacking *WDFY2* were identified by PCR, and the introduced mutations were characterized by PCR followed by cloning and sequencing. qPCR with a primer pair amplifying exons 2–5 (QT00035455, Qiagen QuantiTect) confirmed absence of a *WDFY2* transcript. The following PCR primers were used for characterization: KS741: AAAGCGCATGCGTCCTAGT, KS742: CACCAGGATGCGCATTACAATA).

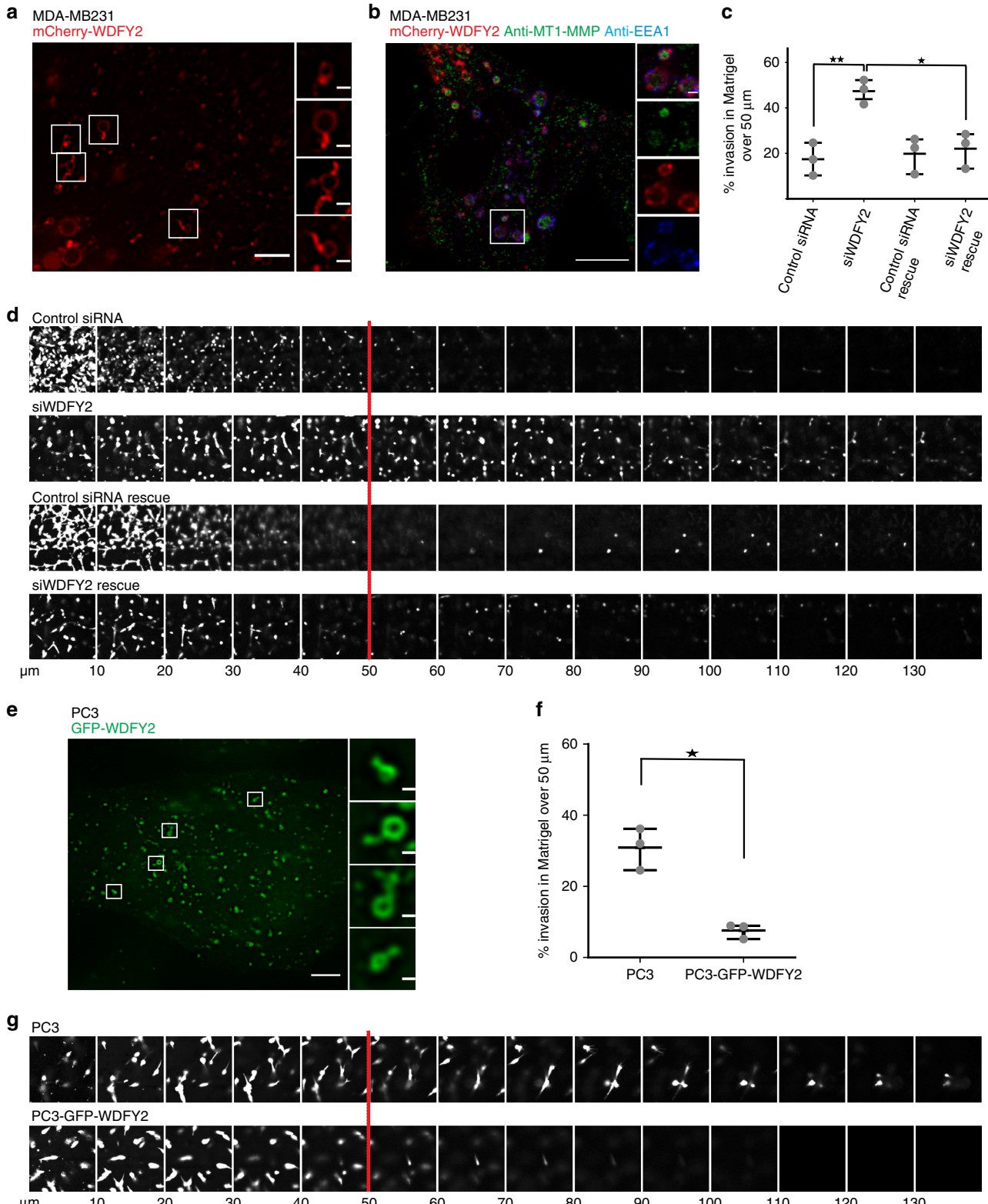

CRISPR–Cas9/AAV-mediated endogenous tagging of WDFY2. hTERT-RPE1 cells expressing endogenous NLAP-WDFY2 fusions were generated by using CRISPR/Cas9 in conjunction with an AAV-based homology donor. The NLAP tagging cassette encodes GFP containing a synthetic intron with a Neomycin resistance cassette, followed by a linker with a PreScission cleavage site, an S-Tag and a TEV cleavage site. A guide RNA binding directly at the start codon (gRNA1: 5′-TGGATCTCCGCGCCATCGG-3′) was cloned into pGuideIt (Clontech). CRISPR/Cas9 was delivered as RNPs using the Clontech Gesicle system (Clontech),

the gesicles were packaged according to the manufacturers manual. To generate the homology donor, a construct containing ~1KB of homology left and right of the start codon and a NLAP tagging cassette[43] was assembled in an AAV vector (pAAV-2Aneo v2, Addgene plasmid 800333) and packaged using pHelper and pRC2-miR321 vectors (Clontech). AAV particles were isolated using AAVPro extraction solution (Clontech). After transduction with both CRISPR/Cas9 gesicles and AAV homology donor, cells with integrated homology donor were selected using Neomycin selection (500 μg/ml). Single clones were picked and characterized

**Fig. 9** WDFY2 controls the invasivity of cancer cells. **a** Deconvolved widefield image showing localization of mCherry-WDFY2 to endosomes and endosomal tubules in MDA-MB231 cells. Representative image from 10 cells. Scalebar: 10 μm, inset: 1 μm. **b** Confocal image of MDA-MB231 cells showing colocalization of MT1-MMP (green) in EEA1 (blue) and mCherry-WDFY2-positive endosomes. Representative image from 8 cells. Scalebar: 10 μm, inset: 1 μm. **c** Quantification of invasion of MDA-MB231 cells transfected with nontargeting siRNA (siControl) and siRNA targeting WDFY2 (siWDFY2) and MDA-MB231 cells expressing siRNA-resistant GFP-WDFY2 transfected with non-targeting siRNA (siControl rescue) and siRNA targeting WDFY2 (siWDFY2 rescue) invading in fibronectin-supplemented Matrigel$^{TM}$. Plotted points indicate the mean invasion derived from 15 z-stacks per experiment, $n = 3$ experiments, shown is the mean ± 95% CI. Anova with Bonferroni post test p value: 0.003. *$p < 0.05$, **$p < 0.01$, ***$p < 0.001$,  n.s. not statistically significant. **d** Optical sections ($\Delta z = 10$ μm) of the conditions described in (**c**), cells stained with Calcein-AM invading into fibronectin-supplemented Matrigel$^{TM}$. The red line indicates the z-axis threshold (50 μm) defining invading cells. **e** Deconvolved widefield image showing localization of WDFY2 to endosomes and endosomal tubules in PC3 cells. Scalebar: 10 μm, inset: 1 μm. **f** Quantification of invasion of  PC3 and PC3 cells stably expressing GFP-WDFY2 cells into fibronectin-supplemented Matrigel$^{TM}$. Plotted points indicate the mean invasion per experiments derived from 15 z-stacks per experiment, $n = 3$ experiments, shown is the mean ± 95% CI. Student's t test, $p = 0.0029$. *$p < 0.05$, **$p < 0.01$, ***$p < 0.001$,  n.s. not statistically significant. **g** Optical sections ($\Delta z = 10$ μm) of PC3 and PC3 stably expressing GFP-WDFY2 stained with Calcein-AM invading into fibronectin-supplemented Matrigel$^{TM}$. The red line indicates the z-axis threshold (50 μm) defining invading cells. Source data are provided as a Source Data file

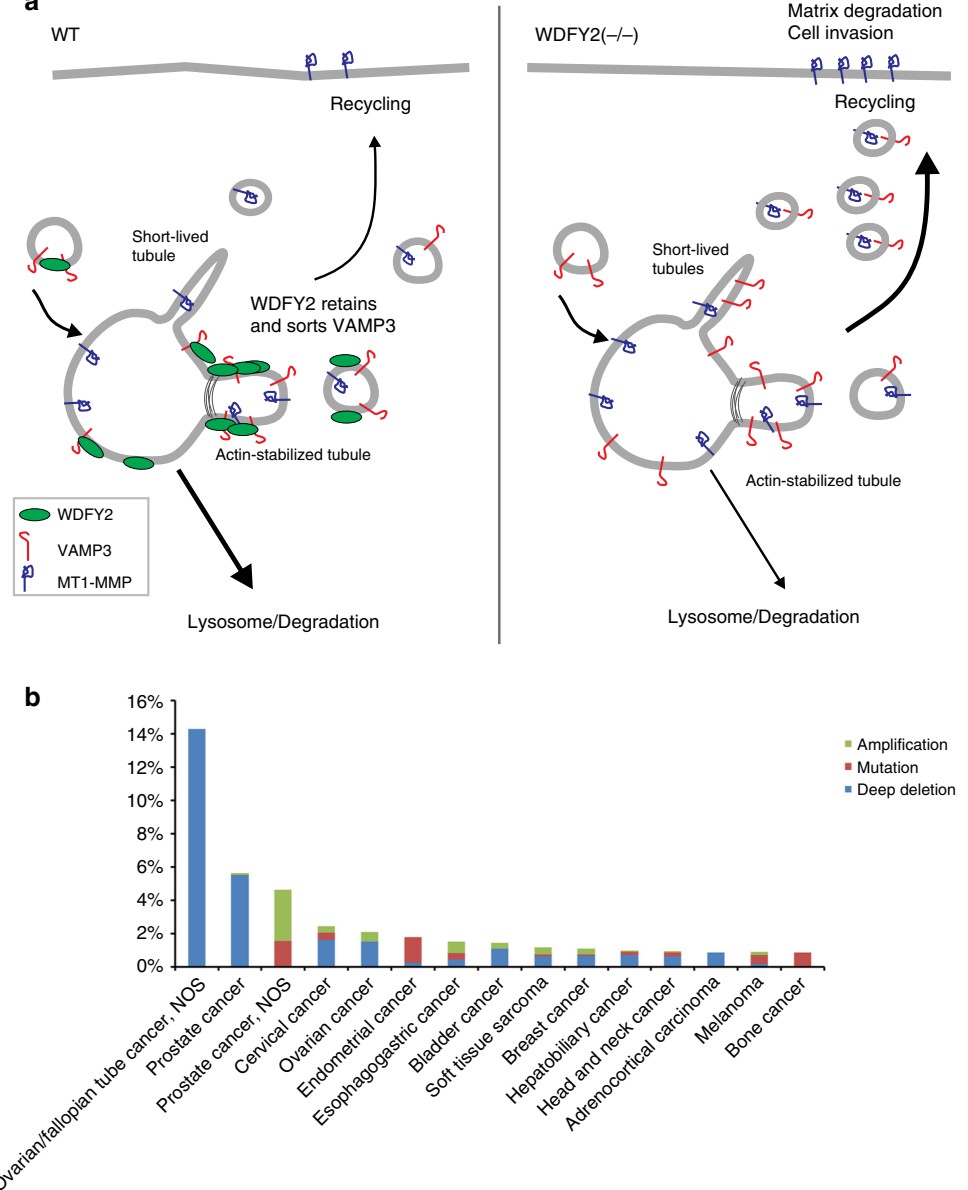

**Fig. 10** WDFY2 controls cancer cell invasion. **a** Model of WDFY2 action in the endocytic system. In WT cells, WDFY2 interacts with VAMP3 and prevents sorting into bulk recycling tubules. This limits the amounts of VAMP3-positive recycling vesicles transporting MT1-MMP. In cells lacking WDFY2, more VAMP3 can be sorted into recycling vesicles, allowing more MT1-MMP to be recycled to the cell surface. **b** WDFY2 is frequently deep deleted in cancer samples. The graph shows data extracted from the CBioportal for Cancer Genomics, alteration frequency is grouped by cancer type and plotted

by PCR and western blotting and the integration site verified by sequencing. The following PCR primers were used for characterization: KS972: GCATGTTGGGA GCAGTAAGC, KS991: CACCACGCCGGTGAACAGTTCC, KS992: CCGGAATCACCCTGGGCATGG, KS975: ACCACGCAGAAGCCTAAACT.

**Confocal fluorescence microscopy**. Confocal images were obtained using LSM710 confocal microscope (Carl Zeiss) equipped with an Ar-laser multiline (458/488/514 nm), a DPSS-561 10 (561 nm), a laser diode 405-30 CW (405 nm), and a HeNe laser (633 nm). Images were taken using a Plan-Apochromat 63×/1.40 oil DIC III (Carl Zeiss).

**Live time-lapse microscopy**. Live-cell imaging was performed on a Deltavision OMX V4 microscope equipped with three PCO.edge sCMOS cameras, a solid-state light source and a laser-based autofocus. Cells were imaged in Live Cell Imaging buffer (Invitrogen) supplemented with 20 mM glucose. Environmental control was provided by a heated stage and an objective heater (20–20 Technologies). Images were deconvolved using softWoRx software and processed in ImageJ/FIJI[26].

**Structured illumination microscopy**. hTERT-RPE1 cells stably expressing GFP-WDFY2 were transiently transfected with mCherry-RAB5/RAB4/RAB11, mCherry-VAMP3 or stained with anti-RAB7/VPS26/VPS35. Cells were fixed with 4% PFA and 0.1% glutaraldehyde and stained with mouse anti-GFP antibody (Roche) and RFP-Booster (Chromotek) to visualize tagged proteins. Alternatively, anti-GFP antibody (Roche) and antibodies targeting RAB7, VPS26, and VPS35 and fitting secondary antibodies were used to visualize endogenous proteins. Samples were mounted in ProLongGold Antifade Reagent (Life Technologies). Three dimensional SIM imaging was performed on Deltavision OMX V4 microscope with an Olympus ×60 NA 1.42 objective and three PCO.edge sCMOS cameras and 488nm and 568nm laser lines. Cells were illuminated with a grid pattern and for each image plane, 15 raw images (5 phases and 3 rotations) were acquired. Super-resolution images were reconstructed from the raw image files, aligned and projected using Softworx software (Applied Precision, GE Healthcare). Images were processed in ImageJ/Fiji[26].

**STORM imaging**. Cells stably expressing GFP-WDFY2 were fixed in 4% PFA and labeled with anti-GFP. Imaging was performed in 100 mM Tris, 50 mM NaCl with an oxygen scavenging system (10% Glucose, 10 kU catalase, 0.5 kU glucose oxidase) and 10 mM MEA as reducing agent. Imaging was performed using a Deltavision OMX V4 microscope (GE Healthcare); localization and reconstruction were performed in softWoRx software; all further image processing was performed in Fiji.

**DNA-PAINT microscopy**. Cells were seeded in an 8-well chamber (Lab-Tek) and fixed with 4% formaldehyde for 15 min in room temperature to preserve endosome tubules. Staining and imaging was performed according to the manufacturers protocol (Ultivue). Imaging was performed using a Deltavision OMX V4 microscope (GE Healthcare); localization, reconstruction and alignment were performed in softWoRx software; all further image processing was performed in Fiji.

**High-content microscopy**. To measure the amount of vesicular MT1-MMP in hTERT-RPE1 and hTERT-RPE1 GFP-WDFY2 cells, we performed high-content microscopy using an Olympus ScanR system with an UPLSAPO 40× objective. Cells were seeded on cover slips and grown to 80–90% confluency before being fixed, permeabilized and stained with an antibody against MT1-MMP. A total of 5 × 5 images were taken at two different places for each cover slip in each experiment, and the total intensity of MT1-MMP positive spots (sized between 5 and 150 pixels) was quantified using the Olympus ScanR analysis program. Identical image acquisition and analysis settings were used for all experiments.

**GFP affinity purification**. Stable hTERT-RPE1 cellines expressing GFP or GFP WDFY2 were seeded in 10 cm dishes up to 80% confluence and then lysed in lysis buffer containing, 50 mM TRIS, 150 M NaCl, 0.25% Triton X100, 1 mM DTT, 50 μM ZnCl$_2$, 5 mM NaPPi, 20 mM NaF, 1× of phosphatase inhibitor 3 (S/T), phosphatase inhibitor 2 (Y), and protease inhibitor mix. GFP-trap magnetic beads (ChromoTek) were added to the lysate and incubated rotating at 4° for 4 h.

**LC–MS/MS, protein identification, and label-free quantitation**. Beads containing bound proteins were washed 3 times with PBS, reduced with 10 mM DTT for 1 h at 56 °C followed by alkylation with 30 mM iodoacetamide in final volume of 100 μl for 1 h at room temperature. The samples were digested over night with Sequencing Grade Trypsin (Promega) at 37 °C, using 1.8 μg trypsin. Reaction was quenched by adding 1% trifluoroacetic acid to the mixture. Peptides were cleaned for mass spectrometry by STAGE-TIP method using a C18 resin disk (3M Empore)[49]. All experiments were performed on a Dionex Ultimate 3000 nano-liquid chromatography (LC) system (Sunnyvale CA, USA) connected to a quadrupole—Orbitrap (QExactive) mass spectrometer (ThermoElectron, Bremen, Germany) equipped with a nanoelectrospray ion source (Proxeon/Thermo). For liquid chromatography separation we used an Acclaim PepMap 100 column (C18, 2 μm

beads, 100 Å, 75 μm inner diameter) (Dionex, Sunnyvale CA, USA) capillary of 25 cm bed length. The flow rate used was 0.3 μL/min, and the solvent gradient was 5% B to 40% B in 120 min, then 40–80% B in 20 min Solvent A was aqueous 2% acetonitrile in 0.1% formic acid, whereas solvent B was aqueous 90% acetonitrile in 0.1% formic acid.

The mass spectrometer was operated in the data-dependent mode to automatically switch between mass spectrometry (MS) and MS/MS acquisition. Survey full scan MS spectra (from $m/z$ 300 to 1750) were acquired in the Orbitrap with resolution $R = 70,000$ at $m/z$ 200 (after accumulation to a target of 1,000,000 ions in the quadruple). The method used allowed sequential isolation of the most intense multiply charged ions, up to ten, depending on signal intensity, for fragmentation on the higher-energy C-trap dissociation (HCD) cell using high-energy collision dissociation at a target value of 100,000 charges or maximum acquisition time of 100 ms. MS/MS scans were collected at 17,500 resolution at the Orbitrap cell. Target ions already selected for MS/MS were dynamically excluded for 45 s. General mass spectrometry conditions were: electrospray voltage, 2.0 kV; no sheath and auxiliary gas flow, heated capillary temperature of 250 °C, heated column at 35 °C, normalized HCD collision energy 25%. Ion selection threshold was set to 1e−5 counts. Isolation width of 3.0 Da was used.

MS raw files were submitted to MaxQuant software version 1.6.1.0 for protein identification[50]. Parameters were set as follow: protein N-acetylation, methionine oxidation and pyroglutamate conversion of Glu and Gln as variable modifications. First search error window of 20 ppm and mains search error of 6 ppm. Trypsin without proline restriction enzyme option was used, with two allowed miscleavages. Minimal unique peptides were set to 1, and false-discovery rate (FDR) allowed was 0.01 (1%) for peptide and protein identification. Label-free quantitation was set with a retention time alignment window of 3 min The Uniprot human database was used (downloaded august 2013). Generation of reversed sequences was selected to assign FDR rates.

**GFP-pulldown assay**. GFP trap (GFP-trap magnetic beads, ChromoTek) was used for interaction studies and the experiments were performed according to the manufacturer's protocol. Stable hTERT-RPE1 cell lines expressing GFP-VAMP3 in combination with mCherry-WDFY2 were used. Stable hTERT-RPE1 cells expressing mCherry-WDFY2 transiently transfected with pEGFP-N2 were used as control. Totally, 5% input was used for the immunoblotting of the GFP traps. All uncropped western blots can be found in Supplementary Fig. 8.

**Protein purification**. GST-tagged WDFY2 was expressed in *E. coli* Rosetta2(DE3) cells (Novagen, Madison, WI). Cells were lysed in lysis buffer (50 mM Tris-HCl, pH 7.5, 100 mM NaCl, 10% glycerol, 10 μM ZnCl2, 2.5 mM 2-mercaptoethanol) and purified using GST sepharose beads (GE Healthcare). Purified protein was eluted in elution buffer (100 mM Tris-HCl, pH 8.0, 100 mM NaCl, 10% glycerol, 10 μM ZnCl2, 2.5 mM 2-mercaptoethanol, and 10 mM reduced glutathione)[51] and directly used in protein–lipid overlay assays.

His-MBP-tagged 2xFYVE domains derived from HRS and WDFY2 were purified by Ni-NTA affinity chromatography. Recombinant protein was expressed in Rosetta2(DE3), purified using HisPur Ni-NTA spin columns (Thermo Fisher) and dialyzed against liposome buffer (50 m Hepes, 150 mM KCL, 100 μM ZnCl2, 1 mM TCEP). Purified protein was flash-frozen in small aliquots and stored at −80 °C.

**Protein–lipid overlay assays**. In vitro lipid-binding activities of full-length WDFY2 were determined by protein–lipid overlay assay[52]. Lipid overlay assays were performed using commercially available PIPStrips (Echelon Biosciences, Salt Lake City, UT) according to the manufacturer's instructions[51].

PIPStrips were blocked for 1 h in blocking solution (1% skimmed milk powder in TBS-T) at room temperature. Next, PIPStrips were incubated overnight at 4 °C with purified GST-WDFY2 (2 μg/ml in blocking solution), washed three times with TBS-T, and incubated with HRP-conjugated anti-glutathione S-transferase (GST) antibody (GE Healthcare) at a dilution of 1:5000 in blocking solution for 1 h. After three washing steps, detection was performed using commercially available ECL reagents (GE Healthcare).

**Liposome flotation assays**. Liposome flotation assays were designed similar to published procedures[21] and performed as follows: Lipids (47% PC, 25% PE, 9% cholesterol, 10% PS, 5% PI (Avanti Polar Lipids), 5% PtdIns3P (Echelon), and 0.2% NBD-PE (Thermo Fisher), all % are molar %) were dissolved in Chloroform. The solvent was evaporated under a nitrogen stream and the lipid film dried under vacuum. The dried lipids were rehydrated in liposome buffer (50 m Hepes, 150 mM KCL, 100 μM ZnCl2, 1 mM TCEP) and multilamellar liposomes were formed by 5 freeze–thaw cycles. The resulting liposome mixture was then extruded by 11 passages through an 800 nm filter membrane. These presized liposomes were then extruded to the final size (400, 200, 100, and 50 nm).

In order to assess curvature-dependent binding, 1 μM recombinant FYVE domains (His-MBP-2xFYVE$_{HRS}$ and His-MBP-2xFYVE$_{WDFY2}$) were added to liposomes (1 mM lipid) in a final volume of 150 μl and incubated for 20 min at RT. To this mixture, 100 μl of 75% sucrose in liposome buffer was added, resulting in a 30% sucrose solution. This fraction was overlaid with 200 μl 25% sucrose in

liposome buffer and 100 µl of liposome buffer without sucrose. The sample was then centrifuged at $240,000 \times g$ in a Beckman swing rotor (TLS 55) for 1 h. Successful liposome flotation was verified by visualizing NBD-PE fluorescence using a Safelight gel imager, and fractions (250 µl (bottom), 200 µl (middle), and 100 µl (top)) were collected from the bottom. Totally, 25 µl of the top fraction was separated using SDS-PAGE and visualized by Coomassie Brilliant Blue staining. Gels were recorded using an Odyssey gel imaging device (LI-COR Biotechnology). Intensities were measured using Fiji and plotted in GraphPad Prism.

**VAMP3 redistribution assay**. hTERT-RPE1 WT and WDFY2(−/−) cells were transiently transfected with GFP-VAMP3 and mCherry-CORTACTIN overnight and fixed in 3% PFA. Cells were labeled with anti-GFP (Roche) and RFP-booster (Chromotek) Images were acquired on LSM710 confocal microscope using a Plan-Apochromat 63×/1.40 oil DIC III (Carl Zeiss) objective. The leading edges or cells were selected in the red channel (CORTACTIN). Images were processed in Fiji using a custom Python script, a schematic of the image processing is shown in Supplementary Fig. 9a. For the generation of superpixels to visualize Vamp3 distribution, cells were outlined using ImageJ and an ImageJ macro (https://gist.github.com/mutterer/035ade419bf9c96475ce) was used to generate hexagonal ROIs. Mean intensity values for each hexagon were measured and the corresponding hexagon filled with the mean value (Supplementary Fig. 9b).

**Measuring compartment-specific VAMP3 localization**. hTERT-RPE1 WT and WDFY2(−/−) cells were transfected with GFP-VAMP3 and fixed in 3% PFA. Cells were labeled with human anti-EEA1 antibody and rabbit anti-Lamp1 antibody and corresponding fluorophore-conjugated secondary antibodies. Images were acquired using a LSM710 confocal microscope using a Plan-Apochromat 63×/1.40 oil DIC III (Carl Zeiss) objective. To measure VAMP3 distribution, EEA1 and LAMP1-positive vesicles were automatically segmented and GFP-VAMP3 intensity was measured within the respective compartments using a custom Python-based Fiji script[26]. Individual measurements were then collected and plotted.

**Phluorin exocytosis experiments**. hTERT-RPE1 WT and WDFY2(−/−) cells were seeded in MatTek dishes (Inter Instruments) and transfected with pHluorin-VAMP3 or pHluorin-MT1-MMP. Imaging was performed on Deltavision OMX V4 microscope (GE Healthcare) using a 60× TIRF objective. Images were taken every second for 2 min. Images were manually processed using Fiji by scoring bright dots that appeared and disappeared within a few frames[26].

**Transferrin recycling assays**. hTERT-RPE1 WT and WDFY2(−/−) cells were incubated with 5 µg/ml Alexa488-Transferrin (Molecular Probes) for 15 min at 37 °C. The cells were then washed and either fixed directly with 4% formaldehyde (timepoint 0) or fixed after additional 10 or 30 min at 37 °C in the presence of unlabeled iron-saturated transferrin (3 mg/ml) (Sigma). The formaldehyde fixed cells were stained with Hoechst33342 (Life Technologies) and analyzed on Olympus ScanR illumination system with an UPLSAPO 40× objective. Wide-field images from random areas at two different coverslips per condition were analyzed by using ScanR software. The total fluorescence intensity per cell of Alexa488-Transferrin was measured by intensity-based segmentation, cells were counted by detection of Hoechst-stained nuclei. Identical imaging and analysis settings were applied for all conditions within one experiment.

**Gelatin degradation assay**. Oregon Green-conjugated gelatin-coated (Life Technologies) coverslips[30] were prepared as follow: coverslips (12 mm diameter, No. 1 thickness, VWR international) were precleaned in 20% nitric acid overnight. After extensive washing, the coverslips were coated with 50 µg/ml poly-L-lysine (Sigma-Aldrich) for 30 min, washed in PBS, and fixed with cold 0.5% glutar-aldehyde (Sigma-Aldrich) in PBS for 15 min on ice. Subsequently, the coverslips were washed in PBS and coated for 20 min with prewarmed 10 mg/mL Oregon Green-conjugated gelatin/2% sucrose in PBS. After coating, the coverslips were washed with PBS and incubated in 5 mg/mL sodium borohydride (Sigma-Aldrich) for 15 min The coverslips were then washed with PBS, sterilized with 70% ethanol, and equilibrated in serum-containing medium for 1 h before the addition of cells. For Gelatin degradation assays, $5 \times 10^4$ WT and WDFY2(−/−) cells were suspended in 1 ml culture medium and added to wells with gelatin-coated cover slips followed by 6 h incubation at 37 °C. Two hour after seeding HGF (100 ng/ml) was added. The cells were then fixed in 3% formaldehyde in PBS for 15 min, permeabilized with 0.1% Triton X-100 (Sigma-Aldrich) in PBS, incubated with Rhodamine phalloidin (Life Technologies) for 15 min and mounted for examination by confocal microscopy. Cells incubated with MMP inhibitor GM6001 (VWR international, J65687.MX) were seeded in growth medium containing the MMP inhibitor. For experiments using siRNA-mediated depletion of MT1-MMP or VAMP3, gelatin degradation assays were performed 48 h after transfection with siRNA.

Samples were analyzed using a LSM710 confocal microscope (Carl Zeiss), a 63× objective and zoom 1.0. cells/field of imaging were chosen on basis of the nuclear staining and gelatin quality, and at least 15 images were randomly taken throughout the cover slips in each experiment, giving at least 120 cells per

experiment for each condition. All images within one experiment were taken with constant gain and pin-hole parameters.

Images were quantified using Image J by running a script to measure the average area of gelatin degradation per cell. Due to variations in the gelatin quality the threshold was determined manually. To account for experimental variation between replicates, measured values were normalized by division of the mean for each individual experiment prior to plotting and statistical analyses.

**Inverted invasion assays in Matrigel™**. Inverted invasion assays[53] were performed as follows: Matrigel™ (Corning) was supplemented with 25 µg/ml fibronectin (Sigma-Aldrich) and 80 µl was added to Transwell® (Sigma-Aldrich, 8 µm pores) filter insets and allowed to polymerize for 45 min at 37 °C. The inserts were then inverted and $4 \times 10^4$ (PC3 cells $6 \times 10^4$) cells were seeded on top of the filter on the opposite side from the Matrigel. The transwells were placed in serum free medium and the upper chamber was filled with serum supplemented medium (10% v/v FBS) with HGF (100 ng/ml). Seventy-two hours after seeding (48 h after seeding in the case of knockdown) cells were stained with Calcein AM (4 µM) (Thermo Fisher) for 1 h before invading cells were visualized with confocal microscopy (Zeiss LSM 710, ×20 objective). Cells that did not make it through the filter were removed with a tissue paper. Section of 10 µm intervals (for quantification) and 1.23 µm intervals (for 3D reconstruction) were captured. Images were analyzed with Fiji[26]. Invasion is presented as the sum of white pixels of all slides from 50 µm and beyond, divided by the sum of white pixels of all slides. A 3D reconstruction of $z$-stacks was done in Paraview (https://www.paraview.org/).

**Inverted invasion assays in collagen**. Inverted invasion assays in collagen were performed and analyzed as described for the inverted invasion assay in Matrigel™. Collagen plugs supplemented with fibronectin were prepared as follows: High concentrated rat tail Collagen (Corning) was added to a tube containing 10× DMEM (Sigma-Aldrich), NaHCO₃ (Merck millipore) and dH₂O and the matrix material was transferred to Transwell filter insets. The pH was measured to be between 7 and 8[53].

**Actin depolymerization assay**. hTERT-RPE1 cells stably expressing GFP-WDFY2 were seeded in MatTek dishes (Inter Instruments). Cells were imaged like described for Live time-lapse microscopy. Images were acquired every 3 s. Cells were imaged for a period before Latrunculin B (Merck, 428020, 10 µM final concentration) or CK666 (Merck Millipore, 100 µM final concentration) was added. Endosomal tubule length before and after treatment was quantified using Fiji.

**PI3K inhibitor experiments**. hTERT-RPE1 cells stably expressing GFP-WDFY2 were seeded in MatTek dishes (Inter Instruments). Cells were imaged like described for Live time-lapse microscopy. Images were acquired every 5 s. Cells were imaged for a period before adding SAR405 (Selleckchem), Wortmannin (Sigma-Aldrich) or DMSO (Sigma-Aldrich) with a final concentration of 6 µM.

**Quantitative real-time PCR of mRNA expression**. mRNA expression analysis was done as described in Pedersen et al.[54]. The primers sets used in the experiment were QuantiTect assays, QT00035455 for WDFY2 and QT00000721 for TBP as reference housekeeping gene (Qiagen).

**Colocalization analysis**. hTERT-RPE1 cells stably expressing GFP-WDFY2 were fixed and stained for RAB4/5/7/11. Images were acquired by confocal fluorescence microscopy with 0.7 µm confocal sections and fixed intensity below saturation. Colocalization was then quantified with Fiji using the JACoP plugin[55]. Manders' colocalization coefficient was used to describe the overlap[56]

**Image processing and data analysis**. All live-cell images were deconvolved using Softworx (GE Healthcare) prior to analysis and presentation. Live-cell images of endogenously-tagged WDFY2 were deconvolved using the ER-Decon-II algorithm[57]. All further image analysis and measurements steps were performed in Fiji using custom Python scripts[26]. Postprocessing of data was performed in Python using the Pandas package, plots were generated using the Seaborn package[58]. All used scripts, as well as all raw data required to reproduce the plots shown in the manuscript are available on GitHub.

**Statistics**. Statistical analysis was performed using Graphpad Prism and the SciPy "Stats" package[59]. Student's $t$ test (two-tailed) was used as a measure for statistical significance in samples with a gaussian distribution. In order to account for differences in staining efficiencies and imaging conditions, experiments involving quantification of intensities and gelatin degradation assays were normalized by division by the mean of the experiment and then analyzed. For analysis of multiple samples, we utilized ANOVA and groups were compared using Bonferroni's Multiple Comparison Test. $*P < 0.05$, $**P < 0.01$, $***P < 0.001$. A table detailing statistical analysis for each graph is supplied as Supplementary Fig. 10. Unless otherwise stated, scatter plots show the mean and 95% confidence intervals.

**Reporting summary**. Further information on research design is available in the Nature Research Reporting Summary linked to this article.

## Data availability

The data that support the findings of this study are available from the corresponding authors (H.S. and K.O.S.) upon request. Data underlying all shown plots and non-cropped western blots are provided as Source Data file. The mass spectrometry proteomics data have been deposited to the ProteomeXchange Consortium via the PRIDE[60] partner repository with the dataset identifier PXD013480.

## Code availability

Python scripts used for image analysis and to reproduce all shown plots are available from GitHub.

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

## Acknowledgements

We are grateful to Philippe Chavrier, Matthew Seaman and Jim Norman for kindly providing plasmid constructs. We thank Knut Liestøl for help with statistical analysis and Camilla Raiborg for critical discussion of the manuscript. We thank Anne Engen for help with cell culture, Chema Bassols for IT assistance, and Eva Rønning and Pepijn Wopken for laboratory assistance. The core facilities for Advanced Light Microscopy and Proteomics at Oslo University Hospital are acknowledged for assistance with microscopes and proteomic analyses, respectively. We thank Trine Håve, Ulrikke Dahl Brinch, Kia Wee Tan and Hélène Spangenberg for assistance with plasmid constructs and characterization of CRISPR clones and Ling Wang for preparing gelatin-coated coverslips. K. O.S. and E.M.H. hold career development fellowships (Nos. 163271 and 162817) and N. M.P a postdoctoral fellowship from the Norwegian Cancer Society (No. 145315). H.S. is supported by grants from the South-Eastern Norway Regional Health Authority (No. 2016087) and the Norwegian Cancer Society (No. 182698). C.C. holds a Young Research Talents grant from the Research Council of Norway (No. 262375). This work was partly supported by the Research Council of Norway through its Centres of Excellence funding scheme, Project no. 262652.

## Author contributions

H.S. and K.O.S. supervised and N.M.P. co-supervised the study. M.S., H.S., and K.O.S. conceived the study. M.S., K.O.S., and N.M.P. designed the experiments. M.S. generated construct, lentivirus, and stable cell lines, performed confocal imaging super-resolution imaging and all live-cell imaging, GFP trap experiment, VAMP3 distribution experiment, the compartment-specific VAMP3 distribution experiment, exocytosis experiment, invasion experiments, Colocalization analysis, PI3K inhibitor experiments, image analysis and quantifications and helped with gelatin degradation assays. K.O.S wrote image and data processing software and analyzed data, performed CRISPR/Cas9 knockouts and endogenous tagging, performed liposome binding assays, generated construct, lentivirus, stable cell lines and helped with STORM imaging. N.M.P performed GFP-pulldown experiments confirming interactions, RT-PCR, High-content microscopy, confocal imaging, Western Blotting, and performed gelatin degradation assays. C.C. performed GFP-trap experiment and analyzed mass spectrometry data. E.M.H, helped with the invasion assay and performed TfR recycling assays. M.S., K.O.S., and H.S. wrote the papert with input from all co-authors.

## Additional information

**Competing interests:** The authors declare no competing interests.

