## [Peer Review File · Nature Communications]

Reviewer #1 (Remarks to the Author):

This study focuses on a yet poorly characterized FYVE domain- and WD40 repeats- containing endocytic protein named WDFY2. Some connection between WDFY2 and the progression of ovarian cancer has been reported and WDFY2 may serve as a tumor-suppressor gene in prostate cancer. However, the mechanism(s) by which WDFY2 contributes to cancer aggressiveness is unknown. First, the authors addressed the distribution of overexpressed GFP-WDFY2 protein in endocytic compartments in RPE1 cells and found some partial co-localization with early endocytic EEA1 protein on a subset of endosomes distant from the plasma membrane. Using counterstaining with several Rab GTPases (mCherry-tagged), the authors observed an association of WDFY2 with some Rab5-negative, Rab4-positive subdomains of the early endocytic compartments. By live cell imaging, GFP-WDFY2 was shown to accumulate at the basis of endocytic membrane tubules. Strikingly, Sneeggen and colleagues observed a segregation of fluorescently tagged tandem repeat of Hrs or WDFY2 FYVE domains with respect to globular vs. tubular domains of the early endosomes, respectively. The authors went on searching for WDFY2 protein partners by GFP-trap IP followed by quantitative MS, which identified the endocytic SNARE protein VAMP3. VAMP3 and WDFY2 were observed on the same endocytic membrane tubules. Additionally, contrasting with perinuclear distribution in WT RPE1 cells, GFP-VAMP3-positive vesicles clustered at the cell periphery in WDFY2 CRISPR/CAS9 KO cells. Peripheral distribution of GFP-VAMP3 vesicles was accompanied by a significant increase of VAMP3-positive vesicle exocytic events, compatible with a negative role of WDFY2 in the recycling of VAMP3-positive vesicles. At the functional level, the authors analyzed the function of WDFY2 in the context of trafficking of MT1-MMP, a trans-membrane matrix metalloproteinase that is key for matrix degradation by cancer cells. Overexpressed fluorescently tagged MT1-MMP was observed in VAMP3-positive vesicles and similar to VAMP3 recycling data, exocytosis of MT1-MMP tagged with pHLuorin was significantly increased upon WDFY2 KO. Additionally, the authors found that degradation of a fluorescent gelatin matrix and invasion of RPE1 cells through Matrigel are increased upon WDFY2 KO. Their conclusion is that WDFY2 exerts a negative control of a VAMP3/MT1-MMP axis, which is required for matrix degradation and tumor cell invasion. In its present stage the study appears preliminary and several aspects must be strengthened in order to support the main conclusions.

1- Association of WDFY2 with various endocytic markers seems complex, especially regarding Rab proteins and Rab subdomains. The degree of colocalization of WDFY2 with different Rab GTPases and Rab subdomains should be better documented (maybe linescans would help) and properly quantified. In addition, localization data are exclusively based on overexpressed proteins, which may affect (subdomain) localization. It seems important to assess WDFY2 localization at the endogenous level?

2- The observation that Hrs and WDFY2 FYVE domains can segregate into distinct endosomal membrane domains is quite puzzling as these FYVE domains have identical phospholipid binding specificity (PI3P). This is an intriguing observation, which possibly underlies the function of WDFY2 in the trafficking of specific cargoes; yet basis for this segregation remains untouched. The possibility that Hrs and WDFY2 FYVE domains have some selectivity for distinct PI3P pools or are influenced by local membrane topological properties such as curvature is intriguing and should be easily tested (for instance see B. Antony's lab pioneering studies). Beside the contribution of lipids, and given established function of endosomal actin in endocytic membrane dynamics and tubulation (and see Fig. 2), the authors may also consider a direct role for F-actin or some cytoskeletal proteins in WDFY2 (and WDFY2 FYVE domain) specific localization (actin disrupting drugs, proximity ligation assay, ...).

3- The authors show a redistribution and increased exocytosis of VAMP3-positive vesicles upon loss of WDFY2 function. It is important to further analyze the morphology (tubular vs. globular) and dynamics of Rab4/Rab5-positive endocytic subdomains in the KO cell line. In addition, in the WDFY2 KO cells, it should also be possible to follow the dynamics of PI3P-positive endocytic subdomains using isolated WDFY2 FYVE domain that should not restore WDFY2 activity. All

together, these experiments should help validating the general model described in Fig. 6b, which otherwise remains largely speculative.

4- The authors show the association of exogenously expressed MT1-MMP in VAMP3-positive, presumably WDFY2-positive vesicles and loss of WDFY2 correlates with increased pHLuorin-MT1-MMP exocytosis. Loss of WDFY2 is also correlated with increased fluorescent gelatin matrix by RPE1 cells, and it is concluded that WDFY2 regulates MT1-MMP ability to degrade the matrix. The effect of WDFY2 KO on gelatin degradation should be properly quantified (only single cells are shown in Fig. 5e). In addition, it is not known whether RPE1 cells express endogenous level of MT1-MMP. Since GM6001 is a generic MMP inhibitor (not specific for MT1-MMP), it is not possible to conclude from Fig. 5e that loss of WDFY2 can increase ECM degradation based on recycling of MT1-MMP (the effect observed in Fig 5e could be due to another MMP that does or does not traffic through the VAMP3 pathway).

5- Data shown in Fig. 6 support the conclusion that loss of WDFY2 converts otherwise non-invasive RPE1 cells into cells with some invasive potential. However, in the absence of a formal proof that WDFY2 affect the recycling of endogenous MT1-MMP (see point 4) it cannot be concluded that WDFY2 affects invasion by controlling a VAMP3/MT1-MMP endogenous axis in RPE1 cells. In addition, it is debated whether invasion through Matrigel involves a-protease (MMP)-dependent invasion program (see Sodek et al BMC Cancer 2008 and discussed in Sabeh et al. J Cell Biol 2009). It cannot be excluded that loss of WDFY2 induces protease-independent ameboid invasive migration through some unknown mechanism. Thus, it would be preferable to assess invasion through a fibrillar collagen network made of polymerized native acid-extracted type I collagen with an absolute requirement for MMP (MT1-MMP) activity for invasion.

Minor Points

1 The authors compare the distribution of GFP-WDFY2 to that of mCherry-Coronin. However, there are several coronin-family members encoded by distinct genes and authors should mention which coronin isoform they used.

Reviewer #2 (Remarks to the Author):

Based on state-of-the-art imaging and proteome analysis, the study by Sneeggen et al. assigns a regulatory endocytic recycling function to the WDFY2 protein. In particular, the authors identified that WDFY2 regulates the secretion of MT1-MMP by VAMP3-mediated endocytic recycling to the plasma membrane. As the authors observed an increase of MT1-MMP secretion and invasion in the absence of WDFY2, the study's outcome suggests why WDFY2 may act as a tumour suppressor, namely by controlling VAMP3-mediated control of MT1-MMP.

Major points

- The authors discuss their findings on the function of WDFY2 in the context of its putative role in tumor suppression e.g. in ovarian cancer. As this study has been carried out in (hTERT-immortalised) epithelial cells from the retina, a replication of their key findings (VAMP3 redistribution, invasion in presence or absence of WDFY2) in tumor cell models would strengthen the authors' conclusions.
- The authors may also extend their study by knocking down PI4K2A in the presence or absence of WDFY2 in order to assess VAMP3 recruitment.

Minor points

- The authors nicely demonstrate the specific binding of WDFY2 to PtdIns3P (Fig. 1), and further employed a PtdIns3P-binding mutant of WDFY2. In order to additionally underline this point the authors may want to utilize classical inhibitors of PtdIns3P production.
- On page 3 (Figure 1) the authors distinguish between WDFY2 localisation and where WDFY2 resides. As there are different pools of WDFY2-positive endosomes detectable (e.g. EEA1 positive

and negative), this notion may be explained with in more detail and clarity.

- On page 6 (Figure 4) the authors define that VAMP3 vesicles localize in an area surrounding the Golgi. VAMP3 vesicles should be counterstained with classic Golgi markers in both presence and absence of WDFY2.

Reviewer #3 (Remarks to the Author):

In this manuscript, the authors have investigated the functional role of WDFY2, a FYVE- and WD40-domain containing protein. They provide evidence that WDFY2 is localized to tubular parts of endosomes, interacts with the v-SNARE VAMP3 and negatively regulates secretion of the matrix metalloprotease MT1-MMP.

This study provides some interesting information on the possible function of WDFY2, in particular the finding that WDFY2 interacts with VAMP3. The model presented in Fig. 7b is appealing. However, it does not rely on very convincing data and this study seems preliminary to me at this stage. My major concerns are the following:

- 1) A key experiment is shown in Fig 5a, b, and movie S5 (sorting of MT1-MMP in WDFY2 tubules and transport in VAMP3 positive vesicles). It is hard to be convinced that it is indeed the case. The authors should document vesicles enriched in VAMP3 and containing VAMP3 arising from tubular parts of the endosomes and moving to the cell surface. In principle, this is doable by live cell imaging.
- 2) None of the immunofluorescence images are quantified, which is not acceptable and makes their interpretation difficult. For instance, what is the percentage of WDFY2 positive tubules that contain MT1-MMP? In Fig. 2, what are the percentages of WDFY2 positive structures that are labelled with coronin, WASH or FAM21?
- 3) It is claimed that following addition of latrunculin, WDFY2 positive tubules start to “hyper-tubulate”. This is not obvious looking at movie S3. Again, tubulation should be quantified. The authors should also investigate the role of actin by other means (i.e. si WASH or the Arp2/3 inhibitor CK666).
- 4) According to Fig. 1b, WDFY2-positive tubules contain Rab4. Rab4 has been associated with fast recycling of MT1-MMP (Frittoli et al, J Cell Biol 2014). On the other hand, the authors provide evidence that WDFY2 is associated with relatively long-lived tubules, which presumably do not recycle transferrin receptor (Puthenveedu et al, Cell 2010). This is confusing to me. Could the authors estimate the percentage of MT1-MMP recycle through the Rab4-dependent pathway compared to the WDFY2 pathway? Does WDFY2 silencing affect transferrin recycling?
- 5) An interesting finding is that Hrs-derived 2xFYVE probe does not seem to localize to endosomal tubular structures in contrast to WDFY2-derived 2xFYVE probe (Fig. 2b). Full length WDFY2 binds only to PI3P by protein-lipid overlay assay. However, this assay is not very specific. Liposome flotation assays with various compositions of phosphoinositides would be more informative.

Response to referees

General comments

First, we would like to thank all 3 reviewers for fair and constructive reviews; their input has allowed us to improve our manuscript substantially. Specifically, we have now addressed the endogenous localization of WDFY2 by generating an endogenously tagged WDFY2 allele. Like overexpressed WDFY2, endogenously expressed WDFY2 localizes to tubular regions of endosomes. New analyses based on this have been added in Figure 1d and 2d. In order to address the differential localization of WDFY2- and HRS-derived FYVE probes, we performed liposome flotation assays and find that WDFY2 shows a preference for membranes with high curvature (new Figure 3) We have used siRNA knockdown and overexpression of WDFY2 in additional cancer models (MDA-MB231 breast cancer and PC3 prostate cancer cells) and find that depletion of WDFY2 in MDA-MB231 cells enhances the invasive potential of these cells, whereas overexpression of WDFY2 in PC3 cells, which have very low endogenous expression of WDFY2, converts these invasive cells into a non-invasive phenotype. New data have been added as new Figure 8. We have performed live-cell imaging of WDFY2-2xFYVE in WDFY2 knockout cells to follow tubulation dynamics in knockout cells and have used PA-mCherry-tagged VAMP3 to follow vesicle dynamics at WDFY2-labelled endosome tubules. These analyses show the co-localization of WDFY2 and VAMP3 on vesicles that bud from endosomal tubules.

Below is our point-by-point response, with referee comments in red.

Referee #1

Major points:

1- Association of WDFY2 with various endocytic markers seems complex, especially regarding Rab proteins and Rab subdomains. The degree of colocalization of WDFY2 with different Rab GTPases and Rab subdomains should be better documented (maybe linescans would help) and properly quantified. In addition, localization data are exclusively based on overexpressed proteins, which may affect (subdomain) localization. It seems important to assess WDFY2 localization at the endogenous level?

We have performed quantification of colocalization using Manders' colocalization analysis and have added these data to the manuscript (new Figure 1c). Due to the lack of functional antibodies against WDFY2, we have generated an endogenously tagged WDFY2 allele and analyzed the localization of this allele by fixed-cell and live-cell imaging (new figures 1d, 2d). Importantly, WDFY2 at endogenous expression shows the same localization as overexpressed WDFY2 to tubular domains of endosomes.

2- The observation that Hrs and WDFY2 FYVE domains can segregate into distinct endosomal membrane domains is quite puzzling as these FYVE domains have identical phospholipid binding specificity (PI3P). This is an intriguing observation, which possibly underlies the function of WDFY2 in the trafficking of specific cargoes; yet basis for this segregation remains untouched. The possibility that Hrs and WDFY2 FYVE domains have some selectivity for distinct PI3P pools or are influenced by local membrane topological properties such as curvature is intriguing and should be easily tested (for instance see B. Antony's lab pioneering studies). Beside the contribution of lipids, and given established function of endosomal actin in endocytic membrane dynamics and tubulation (and see Fig. 2), the authors may also consider a direct role for F-actin or some cytoskeletal proteins in WDFY2 (and WDFY2 FYVE domain) specific localization (actin disrupting drugs, proximity ligation assay, ...).

We thank the reviewer for these helpful comments and have used liposome flotation assays with purified His-MBP-tagged 2xFYVE domains of WDFY2 and HRS with differently sized liposomes to analyze the differential binding preferences. Interestingly, we find that the WDFY2 FYVE domain preferentially binds to smaller vesicles with a higher curvature. This provides a plausible explanation for the preferential localization of WDFY2 to endosomal tubules. These results were added as new figure 3 d,e,f.

After addition of actin disruptin drugs, we can still observe WDFY2 on tubular structures (supplemental figure 2c,d,e), suggesting that actin is not required for the recruitment of WDFY2.

3- The authors show a redistribution and increased exocytosis of VAMP3-positive vesicles upon loss of WDFY2 function. It is important to further analyze the morphology (tubular vs. globular) and dynamics of Rab4/Rab5-positive endocytic subdomains in the KO cell line. In addition, in the WDFY2 KO cells, it should also be possible to follow the dynamics of PI3P-positive endocytic subdomains using isolated WDFY2 FYVE domain that should not restore WDFY2 activity. All together, these experiments should help validating the general model described in Fig. 6b, which otherwise remains largely speculative.

We have analysed the localization of tagged WDFY2-2xFYVE domains and measured length and life-time of endosomal tubules in wild-type and WDFY2(-/-) cells and have added these data as new supplemental figure 5 e,f,g. Cells lacking WDFY2 still show endosomal tubulation and have similar tubule lengths. However, we have observed that the life-time of endosomal tubules is shortened, suggesting that these tubules are either less stable or vesicles are pinched off faster.

4- The authors show the association of exogenously expressed MT1-MMP in VAMP3-positive, presumably WDFY2-positive vesicles and loss of WDFY2 correlates with increased pHluorin-MT1-MMP exocytosis. Loss of WDFY2 is also correlated with increased fluorescent gelatin matrix by RPE1 cells, and it is concluded that WDFY2 regulates MT1-MMP ability to degrade the matrix. The effect of WDFY2 KO on gelatin degradation should be properly quantified (only single cells are shown in Fig. 5e). In addition, it is not known whether RPE1 cells express endogenous level of MT1-MMP. Since GM6001 is a generic MMP inhibitor (not specific for MT1-MMP), it is not possible to conclude from Fig. 5e that loss of WDFY2 can increase ECM degradation based on recycling of MT1-MMP (the effect observed in Fig 5e could be due to another MMP that does or does not traffic through the VAMP3 pathway).

We have quantified gelatin degradation and added this as new figure 7c. To test if the observed effect is mediated by MT1-MMP or other MMPs, we depleted MT1-MMP in WDFY2(-/-) cells by RNAi. This led to a strong suppression of the WDFY2 knockout phenotype, confirming that the observed phenotype was due to MT1-MMP. These data were added as new Figure 7b and 7d.

5- Data shown in Fig. 6 support the conclusion that loss of WDFY2 converts otherwise non-invasive RPE1 cells into cells with some invasive potential. However, in the absence of a formal proof that WDFY2 affect the recycling of endogenous MT1-MMP (see point 4) it cannot be concluded that WDFY2 affects invasion by controlling a VAMP3/MT1-MMP endogenous axis in RPE1 cells. In addition, it is debated whether invasion through Matrigel involves a-protease (MMP)-dependent invasion program (see Sodek et al BMC Cancer 2008 and discussed in Sabeh et al. J Cell Biol 2009). It cannot be excluded that loss of WDFY2 induces protease-independent ameboid invasive migration through some unknown mechanism. Thus, it would be preferable to assess invasion through a fibrillar collagen network made of polymerized native acid-extracted type I collagen with an absolute requirement for MMP (MT1-MMP) activity for invasion.

As suggested, we have performed inverted invasion assays using Collagen. Importantly, also here we observe increased invasion in WDFY2 knockout cells. This is shown and quantified in Figure 7h

Minor Points

1 The authors compare the distribution of GFP-WDFY2 to that of mCherry-Coronin. However, there are several coronin-family members encoded by distinct genes and authors should mention which coronin isoform they used.

We apologize for the omission and have now added this information to the Figure, Figure legends and the Material and Methods.

Referee #2:

• The authors discuss their findings on the function of WDFY2 in the context of its putative role in tumor suppression e.g. in ovarian cancer. As this study has been carried out in (hTERT-immortalised) epithelial cells from the retina, a replication of their key findings (VAMP3 redistribution, invasion in presence or absence of WDFY2) in tumor cell models would strengthen the authors' conclusions.

We have extended our experiments to two cancer cell lines in order to analyze the role of WDFY2 in these cell types. We have depleted WDFY2 in MDA-MB231 breast cancer cells by RNAi and performed inverted invasion assays. Also in this cell system, we find that depletion of WDF2 enhances cell invasion. These results were added as new Figure 8 a-d

We also tested PC3 prostate cancer cells, which have been described to show high levels of MMP-dependent matrix degradation and to have low expression of WDFY2. Interestingly, we found that overexpression of WDFY2 in this cell type strongly reduced 3D invasion into matrigel. These results were added as new Figure 8e-g

• The authors may also extend their study by knocking down PI4K2A in the presence or absence of WDFY2 in order to assess VAMP3 recruitment.

We have tried to inhibit PI 4-kinases using phenylarsine oxide, but we observed massive toxicity and thus have not performed a deeper analysis of the role of PI 4-kinases.

Minor points

• The authors nicely demonstrate the specific binding of WDFY2 to PtdIns3P (Fig. 1), and further employed a PtdIns3P-binding mutant of WDFY2. In order to additionally underline this point the authors may want to utilize classical inhibitors of PtdIns3P production.

We thank the reviewer for this helpful comment and have used SAR405 and Wortmannin to inhibit PI 3-kinases. In line with our proposed model, we observed rapid dissociation of WDFY2 from endosomes upon addition of these inhibitors. The quantification of these results has been added as new Figure 3c

• On page 3 (Figure 1) the authors distinguish between WDFY2 localisation and where WDFY2 resides. As there are different pools of WDFY2-positive endosomes detectable (e.g. EEA1 positive and negative), this notion may be explained with in more detail and clarity.

As suggested, we have clarified this part of the text.

• On page 6 (Figure 4) the authors define that VAMP3 vesicles localize in an area surrounding the Golgi. VAMP3 vesicles should be counterstained with classic Golgi markers in both presence and

absence of WDFY2.

We have added an additional staining with anti-TGN46 and added this as supplemental Figure 5c

Referee #3 :

1) A key experiment is shown in Fig 5a, b, and movie S5 (sorting of MT1-MMP in WDFY2 tubules and transport in VAMP3 positive vesicles). It is hard to be convinced that it is indeed the case. The authors should document vesicles enriched in VAMP3 and containing VAMP3 arising from tubular parts of the endosomes and moving to the cell surface. In principle, this is doable by live cell imaging.

The reviewer raises an important point, and we have now used PA-mCherry-tagged VAMP3 and analyzed its localization on GFP-WDFY2 labelled endosomal tubules. Interestingly, after photoactivation of individual endosomes, we could observe localization of PA-mCH--VAMP3 to WDFY2-labelled endosome tubules and observed the formation of vesicles positive for WDFY2 and VAMP3 from these tubules This is added as new Figure 4d and movie S5. Due to photobleaching, we were not able to follow single vesicles from their formation at the endosome to fusion with the plasma-membrane, but we believe our new photoactivation experiments, in combination with the knockout phenotypes, provide strong evidence for a role of WDFY2 in control of endocytic recycling of VAMP3-containing vesicles.

2) None of the immunofluorescence images are quantified, which is not acceptable and makes their interpretation difficult. For instance, what is the percentage of WDFY2 positive tubules that contain MT1-MMP? In Fig. 2, what are the percentages of WDFY2 positive structures that are labelled with coronin, WASH or FAM21?

We have now added quantification of microscopy data to the individual figures. Specifically, we have added colocalization analysis data to Figure 1c, and quantified tubule localization data as Figure 2h. MT1-MMP localization to WDFY2-labelled tubules was quantified and added in the manuscript text.

3) It is claimed that following addition of latrunculin, WDFY2 positive tubules start to “hyper-tubulate”. This is not obvious looking at movie S3. Again, tubulation should be quantified. The authors should also investigate the role of actin by other means (i.e. si WASH or the Arp2/3 inhibitor CK666).

We have now used CK666 treatment to acutely perturb actin on endosome tubules, and have quantified both the LatB and CK666 data from multiple cells and multiple endosomes per cell (>200 endosomes per condition). This is shown in the new Supplemental Figures S2c and S2D. Moreover, we show now a gallery of endosomal tubules from untreated and treated cells (Suppl. Figure S2e).

4) According to Fig. 1b, WDFY2-positive tubules contain Rab4. Rab4 has been associated with fast recycling of MT1-MMP (Frittoli et al, J Cell Biol 2014). On the other hand, the authors provide evidence that WDFY2 is associated with relatively long-lived tubules, which presumably do not recycle transferrin receptor (Puthenveedu et al, Cell 2010). This is confusing to me. Could the authors estimate the percentage of MT1-MMP recycle through the Rab4-dependent pathway compared to the WDFY2 pathway? Does WDFY2 silencing affect transferrin recycling?

We observed colocalization of WDFY2, MT1-MMP and Rab4 on endosomal tubules (new Supplemental Figure S6 f,g,h and supplemental movies S9,10,11), thus we cannot strictly

distinguish between Rab4- and WDFY2-dependent recycling. It is likely that vesicles derived from WDFY2-labelled tubules enter the Rab4-dependent recycling pathway. We have tested for defects of Transferrin receptor recycling (new figure 5d) and did not observe any changes in WDFY2 knockout cells. This suggests that, while both TfR and WDFY2-sorted cargos use Rab4 for their recycling, their sorting on the endosome is independent of each other.

5) An interesting finding is that Hrs-derived 2xFYVE probe does not seem to localize to endosomal tubular structures in contrast to WDFY2-derived 2xFYVE probe (Fig. 2b). Full length WDFY2 binds only to PI3P by protein-lipid overlay assay. However, this assay is not very specific. Liposome flotation assays with various compositions of phosphoinositides would be more informative.

We have used liposome flotation assays to test for the different binding of WDFY2 and HRS FYVE domains and find that this is based on a preference for different membrane curvatures. This localization is dependent on PtdIns3P, as liposomes with only PtdIns show no binding of the HRS or WDFY2 FYVE domains (data not shown). Moreover, we show that inhibition of PI 3 kinases by Wortmannin or SAR405 leads to a loss of WDFY2 from endosomes.

Reviewers' comments:

Reviewer #1 (Remarks to the Author):

To my opinion, the revised manuscript is improved and authors have done a good job answering some of the points including relationship between WDFY2 and membrane curvature. Yet, the consequences of WDFY2 modulation on VAMP3 and MT1-MMP are studied separately and the conclusion that WDFY2 controls MT1-MMP recycling through regulation of VAMP3 is only correlative. To prove their model, the authors would need to show that changes in MT1-MMP recycling and consequential effects on matrix degradation and invasion due to modulation of WDFY2 expression actually depend on VAMP3. The same lack of causality applies to the title as the data do not prove that VAMP3 retention in tubules due to WDFY2 is the actual mechanism for the effect on MMP secretion. At least these important statements must be toned down.

Reviewer #3 (Remarks to the Author):

The authors have met my previous comments/criticisms in a satisfactory way.

Response to referees

Below is our point-by-point response, referee comments are highlighted in red.

Reviewer #1 (Remarks to the Author):

To my opinion, the revised manuscript is improved and authors have done a good job answering some of the points including relationship between WDFY2 and membrane curvature. Yet, the consequences of WDFY2 modulation on VAMP3 and MT1-MMP are studied separately and the conclusion that WDFY2 controls MT1-MMP recycling through regulation of VAMP3 is only correlative. To prove their model, the authors would need to show that changes in MT1-MMP recycling and consequential effects on matrix degradation and invasion due to modulation of WDFY2 expression actually depend on VAMP3.

We thank the reviewer for raising this valid point. In order to establish a causal link between VAMP3 and WDFY2, we have performed a new set of experiments (new figures 6g and 7e) where we depleted VAMP3 by RNAi in WDFY2(-/-) cells (new Supplemental Figure 6g). We then tested the effect of VAMP3 knockdown on MT1-MMP secretion, measured by pHluorin-MT1-MMP (new Figure 6g), and matrix degradation, measured by degradation of fluorescent gelatin (new Figure 7e). In both cases, we found that depletion of MT1-MMP in WDFY2 knockout cells suppressed the enhanced MT1-MMP secretion and matrix degradation. This demonstrates that the observed defects are indeed dependent on VAMP3 and are not the effect of a VAMP3-independent, WDFY2-dependent process.

The same lack of causality applies to the title as the data do not prove that VAMP3 retention in tubules due to WDFY2 is the actual mechanism for the effect on MMP secretion. At least these important statements must be toned down.

As suggested by the reviewer, we have amended the title, which now reflects our finding that WDFY2 controls VAMP3-dependent recycling of MT1-MMP, without specifically claiming that this is due to a retention in endosomal tubules.